# Development and Characterization of Eco-Efficient Ultra-High Durability Concrete

**Keila Robalo** [1], **Hugo Costa** [2], **Ricardo Carmo** [2,*] and **Eduardo Júlio** [3]

1   Civil Engineering Research and Innovation for Sustainability, University of Cape Verde, Praia 7943-010, Cape Verde
2   Civil Engineering Research and Innovation for Sustainability, SUScita, ISEC—Polytechnic Institute of Coimbra, 3030-199 Coimbra, Portugal
3   Civil Engineering Research and Innovation for Sustainability, Instituto Superior Técnico, University of Lisbon, 1049-001 Lisbon, Portugal
*   Correspondence: carmo@isec.pt

**Abstract:** Ultra-High-Performance Concrete (UHPC) is characterized by having an ultra-compact matrix resulting in ultra-high mechanical properties, low permeability to water and gases, and improved ductility provided by the addition of fibers. The production of structures with this type of concrete is advantageous in some situations, especially in aggressive environments since it significantly increases durability. However, high dosages of Portland cement and silica fume are commonly adopted, increasing not only the cost but also the environmental impact, jeopardizing its use, mainly in the present context where the sustainability of the construction sector is a global priority. In this sense, improving the eco-efficiency of this type of concrete is mandatory. The objective of this work is to develop eco-ultra-high-durability concrete (eco-UHDC). The UHDC matrix was optimized, focusing mainly on durability and looking for the lowest environmental impact, where several parameters were varied: cement replacement ratio, additions in binder matrix and its relative proportions, water/binder ratio, type of fibers, and its proportion. The developed eco-UHDC was characterized both in fresh and hardened states, in terms of mechanical properties, time-dependent properties, and durability. This last topic includes the characterization of durability parameters under laboratory conditions and in a real environment, namely, in the tidal zone of the coast of Cape Verde. The results of resistance to carbonation and chloride penetration were used to predict the service life of structures produced with these eco-UHDC. The optimization of the UHDC matrix allowed the development of mixtures with only 60% of cement in relation to the total amount of powder of the matrix, maintaining good workability and the desired mechanical characteristics (compressive strength higher than 100 MPa and flexural strength higher than 12 MPa). The results also showed that considering only the requirements related to durability, the cover of structures produced with these optimized mixtures can be lower than the values recommended by Eurocode 2, with differences that can reach 55%, mainly when pozzolan of Cape Verde is used as partial replacement of Portland cement.

**Keywords:** ultra-high performance fiber reinforced concrete; durability; sustainability; carbonation resistance; creep and shrinkage

## 1. Introduction

Recently, the environmental impact of the construction industry has been addressed by many researchers, and the concern to reduce it is growing because this industry consumes enormous quantities of non-renewable raw materials and is also one of main contributors of greenhouse gas release. Ultra-High-Performance Concrete (UHPC) can have ultra-high compressive strength, over 150 MPa, combined with ultra-high durability [1]. For example, the chloride diffusion coefficient of a UHPC can be on the order of $2 \times 10^{-12}$ m²/s, significantly lower than that of high-performance concrete, around 4 to $6 \times 10^{-12}$ m²/s [2–7].

However, the huge dosages of Portland cement and silica fume required in its composition increases the economic and environmental costs [3,8–10]. In addition, that level of performance requires special curing conditions, only possible in an industrial environment. Mehta (2009) recommended, according to [3,11], three approaches to improve the sustainability of the concrete industry: (1) consume less concrete, developing innovative architectural and structural projects for both the construction of new structures and rehabilitation of existing structures; (2) reduce the cement Portland dosage in concrete mixes using a smart proportioning approach; and (3) reducing the cement Portland dosage in concrete mixes using a larger volume of one or more additions. The last two approaches are related to the formulation of concrete. Several scientific works have been developed [11–19] with the goal of optimizing the proportion of UHPC constituents, which falls under the second approach. Regarding the third approach, the use of supplementary cementitious materials of micrometric and nanometric size has also been widely studied to partially replace the cement Portland in the production of UHPC, citing as examples the studies carried out by Chan and Chu [20], Droll [21], Tuan et al. [22], Ghafari et al. [4], Zhao and Sun [23], Yu et al. [10], Ferdosian et al. [24], etc.

A typical UHPC mix contains cement Portland, silica fume, quartz powder silica sand, steel microfibers, superplasticizer, and water. Typically, a high proportion of cement is used, between 800 and 1100 kg/m$^3$, approximately three times more than the dosage of current concrete [3]. A wide variety of dosages of microparticles of cementitious materials, mostly pozzolanic or latent hydraulic reaction, has been used to partially replace cement, such as, for example, silica fume, fly ash, ground granulated blast furnace slag, rice husk ash, and metakaolin.

The efficiency of UHPC is particularly dependent on its density, and this can be achieved by optimizing the packing of particles, which consists of an almost perfect granular distribution incorporating a homogeneous gradient of ultrafine, fine, and medium particles in the mixture. In this context, the use of nanometric cementitious materials is highly effective, such as the addition of nano-silica, pozzolanic nanoparticles, nano-metakaolin, and nano-clay [3,25–29]. The studies carried out, considering both the various guidelines on material properties, the UHPC formulation, and the concept of sustainability gave rise to a new family of UHPC, the eco-efficient UHPC. It should be noted that this type of concrete exhibits exceptional mechanical properties and durability but has a significantly lower environmental impact compared to the traditional UHPC formulation.

The present work constitutes an advance in relation to the mentioned studies by improving, even more, the eco-efficiency of the UHPC, which is today accepted to have a compressive strength between 100 and 120 MPa at 28 days [30]. The main objective is to develop eco-ultra-high durability concrete (eco-UHDC), mostly designed with current constituents and with normal curing conditions, where the main property to enhance is not the compressive strength. Therefore, it is not essential to reach a compressive strength of 150 MPa but instead is essential to assure high durability and eco-efficiency, which is why it was named eco-UHDC or eco-UHDFRC (eco-ultra-high durability fiber-reinforced concrete). To enhance the durability of structures and, simultaneously, to reduce the carbon footprint related to cement consumption, the eco-UHDC may be applied only in the cover layer of the structural members where it is most needed to protect the structure, being the core produced with current concrete or even with concrete with significantly lower cement content.

To achieve the mentioned goal, the strategy was to improve the eco-efficiency of the material, minimizing the cement dosage and using more ecological additions to replace it (reduction of about 50% of the cement dosage comparatively to the traditional UHPC formulations), optimizing the dosage of superplasticizer to increase compactness, consequently resulting in a concrete with low porosity. For this purpose, several preliminary mixtures were defined, and, based on the results obtained, the UHDC compositions with the best durability/eco-efficiency ratio and good time-dependent behavior were chosen. The developed eco-UHDC mixtures were characterized in the fresh and hardened state, in

terms of mechanical properties (tensile and compressive strength, Young's modulus, and bending fracture energy), time-dependent properties (shrinkage and creep), and durability (resistance to carbonation, resistance to chloride penetration, chloride content on concrete surface, water absorption by capillarity, and electrical resistivity). Durability parameters were determined using specimens under laboratory conditions and specimens exposed to the maritime environment of Cape Verde, in the tidal zone. The service life of structures related to the risk of reinforcement steel corrosion were also estimated, considering the durability performance of the several eco-UHDC developed, namely, the resistance to carbonation and penetration of chloride ions.

## 2. Experimental Program

### 2.1. Materials

In order to promote a wide range of solutions and parameters that allow defining the formulation strategy, the following constituents were selected for the experimental program: (a) cement type CEM I 52.5R, several current additions (fly ash, limestone filler, silica fume, and quartz flour), and pozzolan from Cape Verde as alternative to fly ash; (b) fine siliceous sand of 0/1 mm size-fraction (granulometry), selected from a preliminary experimental analysis based on three fine sands; (c) three types of microfibers (basalt fibers, glass fibers, and steel fibers, the latter being the traditionally used in UHPC) and one polymeric type of macrofibers; water and superplasticizer (ether–carboxylate based), named M526.

Table 1 and Figure 1 present the main characteristics of the fibers adopted in the study: steel straight microfibers (OL13/0.16); hybrid polymeric macrofibers (S25), constituted by polypropylene and polyethylene; alkali-resistant glass microfibers; and basalt microfibers.

**Table 1.** Characteristics of fibers according to the data sheet of companies.

|  | Steel Fibers | Polymeric Fibers | Glass Fibers | Basalt Fibers |
|---|---|---|---|---|
| Tensile strength | 3000 MPa | 650 MPa | 1700 MPa | 3500 MPa |
| Young's modulus | 200 GPa | 5 GPa | 72 GPa | 95 GPa |
| Density | 7.85 g/cm$^3$ | 0.92 g/cm$^3$ | 2.68 g/cm$^3$ | 2.67 g/cm$^3$ |
| Diameter | 0.16 mm | 0.3 to 0.5 mm | 14 μm | 18 μm |
| Length | 13 mm | 25 mm | 12 mm | 13 mm |

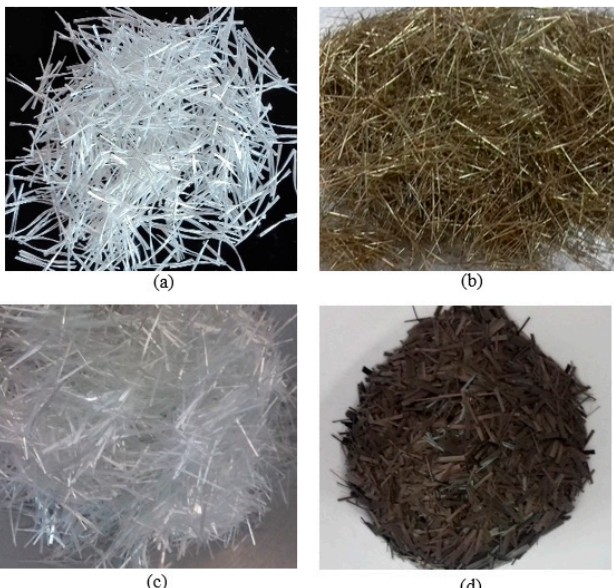

**Figure 1.** Fibers used in the formulation of eco-UHDC: (**a**) polymeric fibers, (**b**) steel fibers, (**c**) glass fibers, and (**d**) basalt fibers.

### 2.2. Formulation and Optimization of Eco-UHDC

Once the properties of the constituent materials of the various UHDC mixtures were characterized, several preliminary mixtures were developed to calibrate some parameters of the formulation, aiming to optimize the binder paste by combining the intended performance level with the reduction of the ecological and economical costs. It should be noted that, at that first stage, the goal was to evaluate the performance of UHDC matrices without reinforcing fibers. It is usual for the powder dosage in the matrix to be about 1200 to 1400 kg/m$^3$, and in this study, the dosage of the powder in the binder paste was fixed at 1100 kg/m$^3$, allowing the addition of high fiber content, maintaining good workability. The target air content was set at 1.0% to 1.5%. Table 2 shows the different formulated mixtures. The mixture proportioning of materials was performed by weight, considering the design parameters used in the literature regarding W/C ratio and W/B ratio, among others, and optimizing the packing density.

**Table 2.** Dosages of constituents of the developed eco-UHDC.

| Mixtures | | C = 75%; FA = 15%; Other Additions = 10% | | | C = 60%; LF = 25%; Other Additions = 15% | | | | C = 60%; LF = 15%; FA =25% |
|---|---|---|---|---|---|---|---|---|---|
| | | SK1 | SK2 | SK3 | SK4 | SK5 | SK6 | SK7 | SK8 |
| Binder powder dosage | kg/m$^3$ | | | | 1100 | | | | |
| Water/binder powder (W/B) | | 0.195 | | 0.18 | | | | 0.21 | |
| Cement (C) | (kg/m$^3$) | 825 | 825 | 660 | 660 | 660 | 660 | 660 | 660 |
| Limestone filler (LF) | (kg/m$^3$) | 110 | | 275 | 275 | 275 | 275 | 275 | 165 |
| Fly ash (FA) | (kg/m$^3$) | 165 | 165 | 165 | 165 | | | | 275 |
| Silica fume | (kg/m$^3$) | | | | | | 165 | | |
| Pozzolan—Cape Verde | (kg/m$^3$) | | | | | | | 165 | |
| Quartz flour | (kg/m$^3$) | | 110 | | | 165 | | | |
| Water | (kg/m$^3$) | 215 | 215 | 198 | 231 | 231 | 231 | 231 | 231 |
| Admixture (Spl) | (kg/m$^3$) | 29 | 35 | 30 | 13 | 15 | 25 | 18 | 17 |
| Sand 0/1 mm | (kg/m$^3$) | 983 | 969 | 1006 | 961 | 974 | 919 | 937 | 936 |

Several additions were used in the formulation, as mentioned. The strategy was defined by considering two levels of cement content (825 and 660 kg/m$^3$, corresponding to 75% and 60% of the powder content), combined with micro-filler powders (limestone or quartz flour) to assure filler effect and with other pozzolanic materials (fly ash, silica fume, or natural ground pozzolan). Preliminary tests proved that lower cement dosages result in higher risk of segregation to assure proper flowability, where the mechanical performance is also affected. Regarding the water/(binder powder) ratio (W/B), which was defined as 0.21 for the mixtures with 60% of cement as powder content, except SK3 where 0.18 ratio was used; however, this parameter requires, together with the mixtures with SK1 and SK2 (cement corresponding to 75% of the powder content and W/B of 0.195), higher superplasticizer (Spl) dosage (between 29 and 35 kg/m$^3$). As known, silica fume also requires high proportions of superplasticizer compared to the fly ash and pozzolan additions. The superplasticizer proportion was adjusted for each mixture to assure similar flowability and air content between mixtures. Mixture SK8 was formulated with similar parameters to those of SK4, changing the corresponding proportions of fly ash and limestone filler.

Based on the results of tests carried out both in the fresh and in the hardened states presented in the following section, and taking into account mainly the eco-efficiency (which in this case is simply considered the minimum cement dosage), SK4 was selected for the later phase of the study where different fiber types were added, resulting in various fiber-reinforced eco-UHDC mixtures, named eco-UHDFRC: mixtures with 1 and 2% of steel fibers; mixtures with 1 and 2% of basalt fibers; mixtures with 1 and 2% of glass fibers; mixtures with 2 and 4% of polymeric fibers; mixtures with 1% of glass fibers and 2% of polymeric fibers (Table 3). The dosage of fibers in the mixture was defined based

on the literature to avoid the workability loss of the concrete, considering the difficulties in mixing and in the lack of flowability caused by the agglomeration of fibers when a certain proportion of fiber content is exceeded. Depending on the variation of the type or quantity of fibers, small adjustments were made, always necessary, mainly in the dosage of superplasticizer and in the air content to approximate the measured values to target values initially defined, assuring objective and coherent further analysis. Small adjustments of sand proportions resulted mainly in those variations, depending also on the constituent's density.

**Table 3.** Dosages of constituents of the developed eco-UHDFRC.

| Mixtures | C | LF | FA | Water $(kg/m^3)$ | Spl | Sand 0/1 mm | Fibers Types and % by Volume | $(kg/m^3)$ |
|---|---|---|---|---|---|---|---|---|
| SK4s1 | | | | | 13 | 934 | Steel—1% | 79 |
| SK4s2 | | | | | 13 | 907 | Steel—2% | 157 |
| SK4b1 | | | | | 18 | 920 | Basalt—1% | 27 |
| SK4b2 | | | | | 21 | 888 | Basalt—2% | 53 |
| SK4p1 | 660 | 275 | 165 | 231 | 13 | 907 | Polymeric—2% | 18 |
| SK4p2 | | | | | 14 | 853 | Polymeric—4% | 37 |
| SK4g1 | | | | | 18 | 920 | Glass—1% | 27 |
| SK4g2 | | | | | 21 | 874 | Glass—2% | 53 |
| SK4gp | | | | | 18 | 868 | Glass—1% + Polymeric—2% | 27 + 18 |

*2.3. Fresh State and Mechanical Properties*

After the formulation, the production and characterization of eco-UHDC mixtures were performed. In the fresh state, the eco-UHDC mixtures were characterized regarding flowability by the mini-slump flow test according to EN 1015-3 [31] and the density and the air content by the pressure gauge method according to EN 1015-6 [32] and EN 1015-7 [33], respectively.

The eco-UHDC mixtures were also characterized in a hardened state. Tests were performed to access flexural and compressive strength on mortars performed on prismatic specimens (40 mm × 40 mm × 160 mm) according to EN 1015-11 [34], and the results were obtained by averaging three specimens for flexural and six half specimens for compressive at 7, 28, and 56 days of age. After calibrating the mixtures, those presenting higher strengths, higher eco-efficiency (lower cement content), and good workability were selected. Those selected mixtures were characterized in terms of the remaining relevant mechanical properties, such as Young's modulus and fracture energy in bending. The average value of the Young's modulus of eco-UHDC was determined according to the specification LNEC E-397 [35], adjusted to prismatic specimens 50 mm × 50 mm × 300 mm. The tensile fracture energy of the UHDFRC was determined according to EN 14651 [36] to evaluate the contribution of the fibers in tension. For this purpose, two prismatic specimens 150 mm ×150 mm × 600 mm were used, with a notch at mid-span 25 mm high and 5 mm wide on the side parallel to the casting direction (Figure 2). The three-point bending test was performed at a speed rate of 3 mm/s; the distance between supports was 500 mm and four LVDTs (Linear Variable Differential Transformer) were used, two placed at mid-span on opposite faces, and the other two aligned to the supports, as shown in Figure 2.

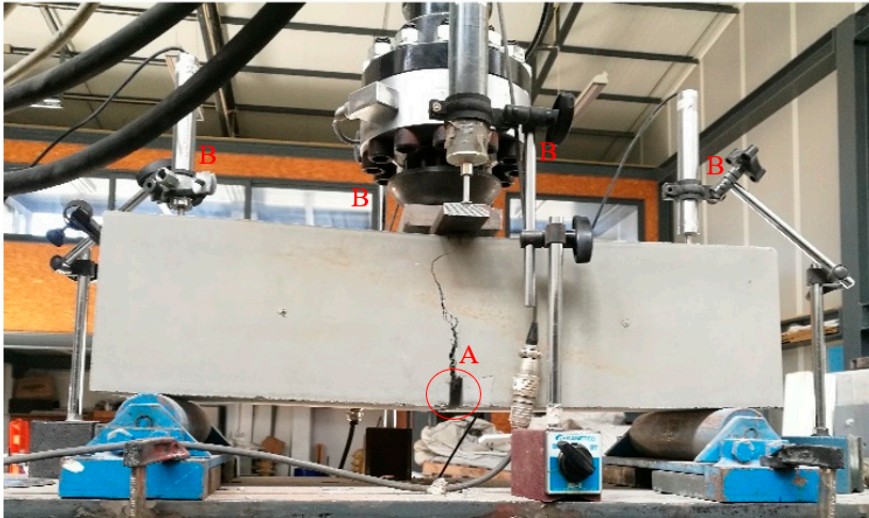

A – Notch
B – LVDTs

**Figure 2.** Test set-up and instrumentation.

### 2.4. Time-Dependent Tests

The time-dependent properties of selected eco-UHDC were characterized. The shrinkage of the selected eco-UHDC was performed experimentally according to the procedure described in EN 12808-4 [37] using prismatic specimens 40 mm × 40 mm × 160 mm. During the casting, metallic inserts were placed on the ends of the specimens to facilitate the shrinkage measurements. The specimens were demolded 24 h after casting and their initial lengths were recorded. During the test, the specimens were kept in the thermo–hygrometric chamber set at a temperature T = 20 °C (±2 °C) and relative humidity RH = 50 % (±5%). The shrinkage test was performed only on the eco-UHDC (without fibers) since SK4 has the same binding matrix as the fiber-containing concrete, and it was intended to evaluate the influence of the paste parameters on the shrinkage of the matrix (without fibers). In addition, it is known that steel fibers tend to significantly reduce the shrinkage of the concrete because the shrinkage deformation is restricted by the fiber-matrix bonds and fibers also give rise to a more cohesive and bonded matrix, mitigating the deformation [38–40].

The creep was also characterized experimentally through the compression creep test according to the specification LNEC E-399 [41], using prismatic specimens 100 mm × 100 mm × 400 mm. The specimens were loaded at 28 days and subjected, throughout the test period, to curing in a thermo–hygrometric chamber set at temperature T = 20 °C (±2 °C) and relative humidity RH = 50% (±5%). The creep is a time-dependent property of concrete, characterized by increasing strains over time at a constant stress loading level. This increase is a result of the viscoelastic behavior of the material itself. This property is essentially affected by the hydrated binder matrix and by the aggregates, which mainly have a restriction role [42,43]. Creep is greatly affected by the Young's modulus of the aggregates used in concrete production. Aggregates with higher Young's modulus provide greater opposition to creep deformation of the binder matrix [44]. The creep coefficient of UHPC is usually lower than that of ordinary concrete. Cement type, its dosage, and water/binder ratio significantly affect concrete creep [45], but this relation is not linear, as creep depends on the amount of hydrated binder in the matrix. Therefore, the evolution of the creep coefficient over time, $\varphi(t)$, was evaluated only for SK7 (mixture with pozzolan addition of Cape Verde and W/B = 0.21) and SK4 (mixture with fly ash addition and W/B = 0.21) to determine which of the two additions is the best option to replace the cement, considering the creep.

### 2.5. Durability Tests

The selected eco-UHDC mixtures were submitted to laboratory tests, aiming to evaluate the performance relative to durability: resistance to carbonation, resistance to chloride

penetration, chloride content on the concrete surface, water absorption by capillarity, and resistivity. These tests were performed as described in the scheme shown in Figure 3.

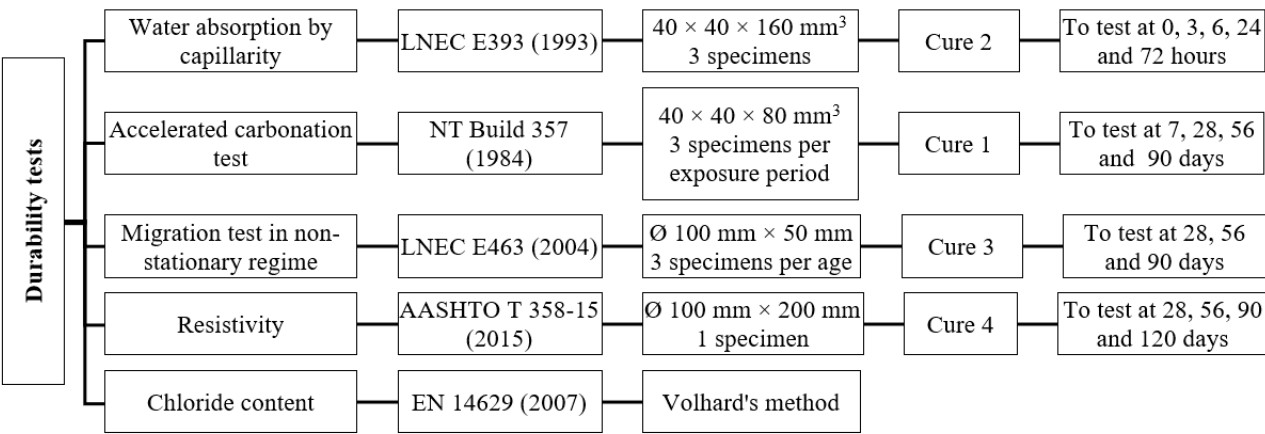

Cure 1 – 14 days of water cure (T = 20 °C ± 2 °C); 14 dry cure days (HR = 50 ± 5 % e T= 20 °C ± 2°C); Cure 2 – 28 days of water cure (20 °C ± 2 °C); Cure 3 – 7 days of water cure (T = 20 °C ± 2 °C); 21 dry cure days (HR = 50 ± 5 % and T = 20 °C ± 2 °C); Cure 4 – curing in water (20 °C ± 2 °C) throughout the test period.

**Figure 3.** The durability tests performed.

The determination of water absorption by capillarity was determined only in SK4 (eco-UHDC with fly ash) and SK7 (eco-UHDC with pozzolan from Cape Verde) to analyze the influence of the two additions on the water absorption mechanism by capillarity. The eco-UHDC reinforced with steel fibers and polymeric fibers were only characterized regarding resistance to carbonation, resistance to chloride penetration, and resistivity, aiming to evaluate the influence of the incorporation of the fibers in these parameters. It was not considered relevant to study the absorption of water by capillarity because the matrix of these concretes is the same as that of SK4 and, according to Neves and Gonçalves [46], the fibers have a negligible influence on the absorption of water by capillarity.

The chloride penetration resistance and the resistivity test were also carried out, both in eco-UHDC (SK4) with and without fibers, as well as in SK3, SK8, and SK7 mixtures, aiming to evaluate the influence of increasing the dosage of fly ash, decreasing of the water/binder ratio, and the replacement of fly ash by pozzolan from Cape Verde in these performance indicators.

The chloride ion content on the concrete surface (Cs) was determined to assess the risk of corrosion of structures produced with eco-UHDC, more specifically with SK7 and SK4 mixtures. This was performed using the second halves of the non-steady state chloride migration test specimens that were not sprayed with the silver nitrate solution. A portion parallel to the edge of the surface that was exposed to the chloride solution, approximately 5 mm deep or less, was removed from these concrete specimens; this value depended on the results of the depth of penetration of chloride ions.

In addition to the durability tests in a laboratory environment, durability tests were also carried out in real environmental conditions, namely in the tidal zone of Cape Verde coast between July 2018 and January 2021. For this purpose, the following eco-UHDC series SK4 were used: SK4 without fibers, SK4s2 with steel fibers, and SK4p2 with polymeric fibers. For each concrete, two cubic specimens with 100 mm edges were produced. These specimens were subjected to wet curing for 28 days and were subsequently placed in the tidal zone. Chloride ion penetration was measured in all directions since none of the faces were waterproofed. The specimens, after exposure, were divided in half, and each half was used to perform a specific test: one to determine the penetration depth of chlorides and the other to determine the chloride content.

## 3. Results and Discussion

### 3.1. Preliminary Mixtures

#### 3.1.1. Properties in Fresh State

The results of the mini-slump flow test and air content in the eco-UHDC mixtures are shown in Table 4. The eco-UHDC had a spread between 300 and 350 mm, presenting good workability and high flowability. The mixtures with higher cement dosage have greater workability and fluidity. Quartz flour and fly ash were the additions that provided better flowability due to the spherical shape of their particles. On the other hand, it was found that to meet the workability requirements in the mixtures with the remaining additions, it was necessary to increase the dosage of superplasticizer, the mixture with silica fume being the one that required the greatest amount of superplasticizer (approximately 25 kg/m$^3$, which is about 92% higher than the mixture with fly ash).

**Table 4.** Flow table and air content test results of eco-UHDC and eco-UHDFRC.

|  |  | Air Content (%) | Slump Flow Spread (cm) |
|---|---|---|---|
| Eco-UHDC | SK1 | 1 | 34 |
|  | SK2 | 0.9 | 35 |
|  | SK3 | 1.1 | 31 |
|  | SK4 | 1.5 | 33 |
|  | SK5 | 0.6 | 34 |
|  | SK6 | 1.7 | 30 |
|  | SK7 | 1.5 | 32 |
|  | SK8 | 1.2 | 35 |
| Eco-UHDFRC | SK4s1 | 1 | 33 |
|  | SK4s2 | 1.2 | 32 |
|  | SK4b1 | 1.5 | 30 |
|  | SK4b2 | 1.9 | 24 |
|  | SK4p1 | 2.8 | 29 |
|  | SK4p2 | 3.1 | 25 |
|  | SK4g1 | 1.4 | 28 |
|  | SK4g2 | 2 | 26 |
|  | SK4gp | 1.5 | 27 |

Basalt and glass fibers are ultra-fine and have a high density relative to the number of fibers in the matrix, which impairs workability and the release of air in the matrix. Those fibers provide higher stability to the mixtures but decrease the flowability and, consequently, the air content can increase up to circa 3%. The polymeric fibers have larger dimensions and allow an addition of greater fiber volume without significant loss of workability.

#### 3.1.2. Compressive and Tensile Strength of Eco-UHDC

The average results of the compressive ($f_{cm}$) and tensile strength ($f_{ctm}$) of eco-UHDC obtained through the test of three identical specimens for each age are shown in Figures 4 and 5.

Analyzing the graphs in Figures 4 and 5, it is possible to verify that all eco-UHDC matrices show an evolution, with age, of the compressive and flexural strengths. The eco-UHDC mixtures with 660 kg/m$^3$ of cement present average values of compressive strength at 56 days between 92 and 116 MPa, and the tensile strength is between 12 and 16 MPa. The compressive and flexural strengths of SK1 and SK2 were not determined. The presented results demonstrate the potential for optimizing the production of eco-UHDC with a lower dosage of cement because the mixtures with only 60% cement present similar performances to the mixtures with 75% of cement. The literature focuses this point, as the UHPC must have a reduced W/B ratio (between 0.15 and 0.25), but with the reduction of cement dosage, compared to traditional formulations, the W/C ratio can have a higher level without impairing the performance of the concrete. The mixtures that obtained the

highest strengths and simultaneously had the lowest amount of cement are SK3 (mixtures with 60% cement, 25% limestone filler and 15% fly ash and W/B = 0.18), SK4 and SK6 (mixtures with W/B = 0.21, 60% cement, 25% limestone filler, and the remaining 15% fly ash, and silica fume, respectively in SK4 and SK6), and SK8 (mixtures with 60% cement, 15% limestone filler, 25% fly ash and W/B = 0.21). It is quite evident the strength evolves with age due to the pozzolanic effect in several mixtures.

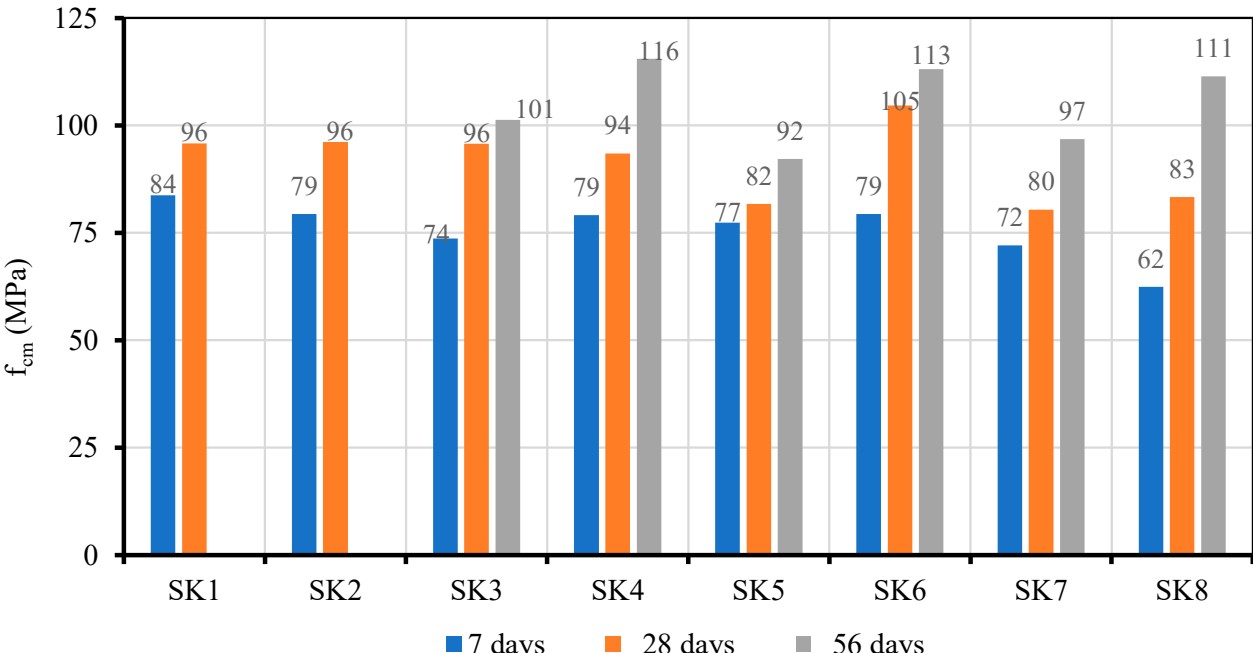

**Figure 4.** Average compressive strengths of eco-UHDC at 7, 28, and 56 days.

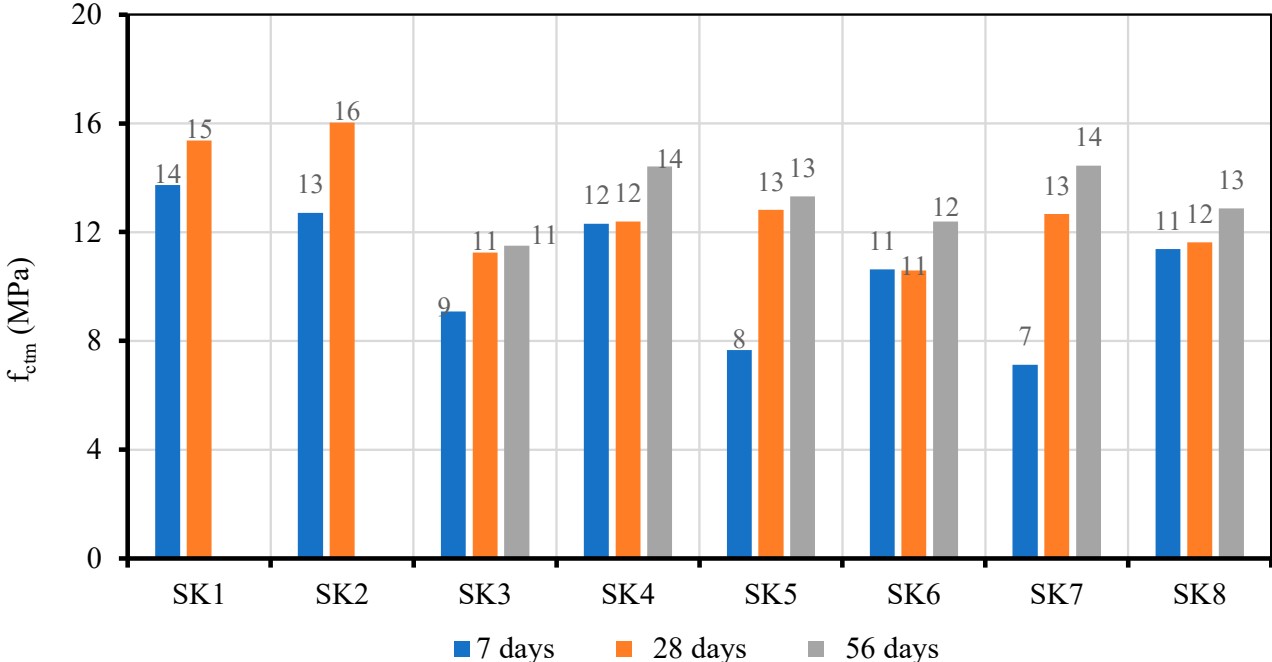

**Figure 5.** Average flexural strengths of eco-UHDC at 7, 28, and 56 days.

### 3.1.3. Compressive and Tensile Strength of Eco-UHDFRC

Figures 6 and 7 present the average compressive ($f_{cm}$) and tensile strength ($f_{ctm}$) of eco-UHDFRC for each of the ages.

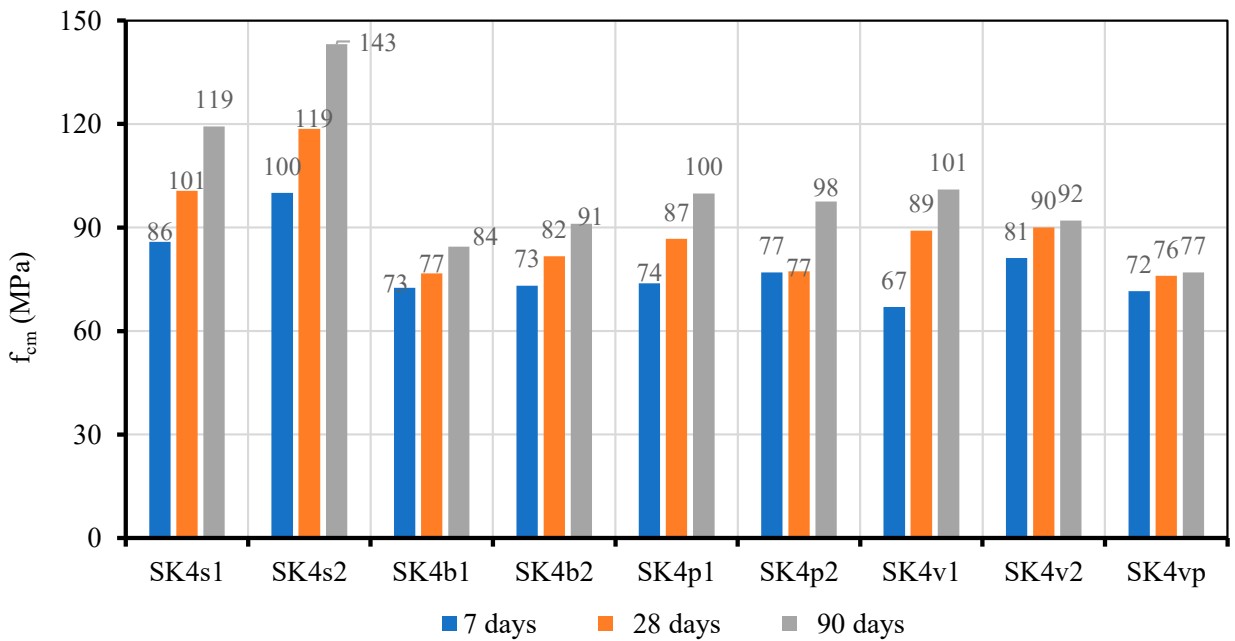

**Figure 6.** Average values of eco-UHDFRC compressive strength at 7, 28, and 90 days.

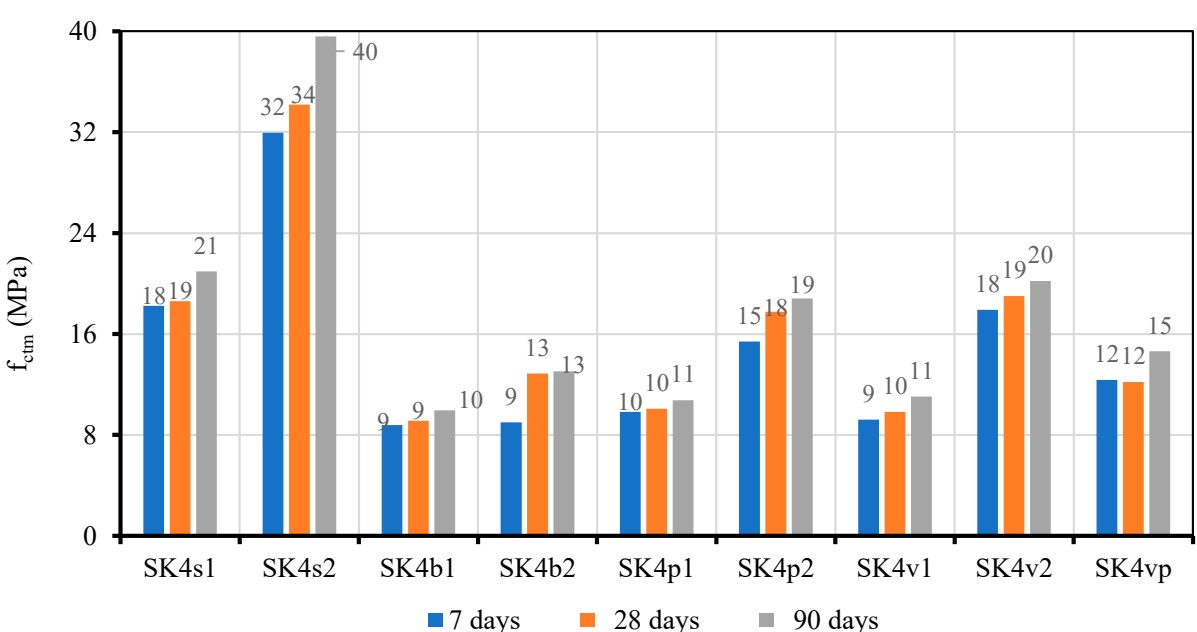

**Figure 7.** Average values of eco-UHDFRC tensile strength at 7, 28, and 90 days.

The results presented in Figures 6 and 7 demonstrate the high-performance potential of steel fibers compared to basalt, glass, and polymeric fibers. The eco-UHDC with steel fibers present compressive strength at 28 days of age 31 to 45% higher than those of eco-UHDC with basalt fibers, and between 12 and 32% higher than those of eco-UHDC with glass fibers, these differences being dependent on the fiber addition rate (1 and 2%, respectively). Concerning tensile strength, the eco-UHDC with steel fibers at 28 days present higher values compared to eco-UHDC with basalt fibers, around 104% in mixtures with 1% fiber and 166% in mixtures with 2% fiber, and in relation to eco-UHDC with glass fibers, the

difference is 89% in the mixtures with 1% of fibers and 80% in the mixtures with 2% of fibers. The eco-UHDC with 2% of polymeric fibers at 28 days presents a compressive strength 14 and 27% lower than the eco-UHDC with 1% and 2% of steel fibers, respectively.

Regarding tensile strength, the eco-UHDC with 1% and 2% steel fibers present, respectively, values 85% and 240% higher than eco-UHDC with 2% polymer fibers. Compared to the eco-UHDC reference, SK4, steel fibers caused an increase of approximately 7 to 27% in compressive strength at 28 days and 50 to 176% in tensile strength, depending on the fiber addition rate.

Comparing the performance of eco-UHDC with basalt, glass, and polymeric reinforcement fibers, it is noted that, given the parameters analyzed, there are no significant improvements compared to the reference eco-UHDC, SK4, and there are even mixtures that show strength losses. Basalt and glass fibers have high strength but are ultra-fine fibers and have a high density toward the number of fibers in the matrix, which impairs workability and the air release in the matrix, resulting in less compact mixtures, less rigidity, and less compressive strength. The failure is brittle, not promoting any increase of ductility (Figure 8). In the case of polymeric fibers, larger dimensions allow the addition of a greater volume of fibers without a significant loss of workability; however, on the other hand, it promotes a reduction of the homogeneity of the matrix, with a consequent loss of compressive strength. The flexural strength of UHDC with polymeric fibers does not increase as much as with steel fiber reinforcement but provides good ductility (Figure 8).

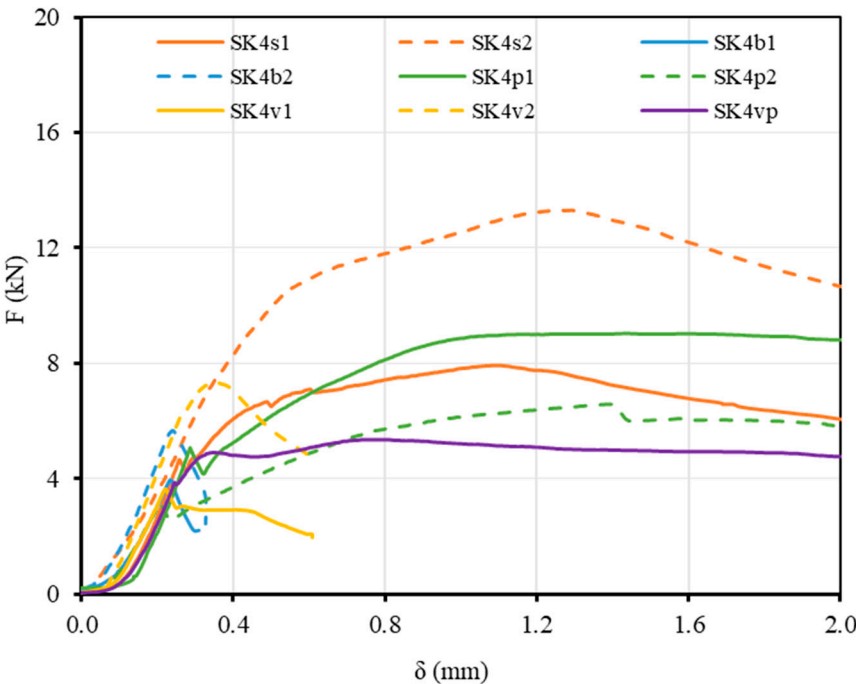

**Figure 8.** Load-deflection curves of the flexural test of eco-UHDFRC at 28 days.

### 3.2. Optimized Eco-UHDC Mixtures

Based on the mechanical properties of the results presented in the previous section, four eco-UHDC mixtures, SK3, SK4, SK7, and SK8, and two eco-UHDFRC mixtures, SK4s2 and SK4p2, were selected. These mixtures were characterized in more detail, namely in the remaining relevant mechanical properties (Young's modulus, flexural fracture energy), time-dependent properties (shrinkage and creep), and durability performance (capillary absorption; resistances to chloride and to carbonation; electrical resistivity).

SK3 was chosen to evaluate the reduction of the water/cement ratio in the main properties, remembering that this concrete differs from SK4 only in this ratio, and in small adjustments of the superplasticizer and of the volume of sand. The SK8 was chosen to evaluate the influence of the fly ash increase, differing from the SK4 only in the amount

of fly ash, which is 10% higher. The SK7 mixture was chosen to evaluate the effect of Cape Verde pozzolan, compared to the SK4 concrete with fly ash. The SK4s2, eco-UHDC with 2% steel fibers, was selected because it presented better results compared to the other UHDFRC and presented significant improvements in mechanical performance compared to the reference matrix without fibers (SK4). Regarding tensile strength, the mixture with steel fibers reached a strength level that was almost triple the reference mixture at 28 days and with a 27% increase of the compressive strength. On the other hand, SK4p2, UHDC with 4% polymeric fibers, was selected due to the ductility showed after cracking (Figure 8).

### 3.2.1. Young's Modulus

The Young's modulus of concrete depends on the stiffness of its main components, i.e., the binder matrix, aggregates, and fibers. All the optimized mixtures were submitted to this test to evaluate the influence of the binder matrix variation, the water/cement ratio, and the type and dosage of additions (comparing SK4 with SK3, SK7, and SK8) on the Young's modulus. Comparing SK4, SK4s2, and SK4p2, it is possible to evaluate the influence of steel and polymeric fibers with the maximum dosages used on the mentioned property. Figure 9 shows the values obtained experimentally for the Young's modulus ($E_{cm}$) of the optimized eco-UHDC at 28 days of age.

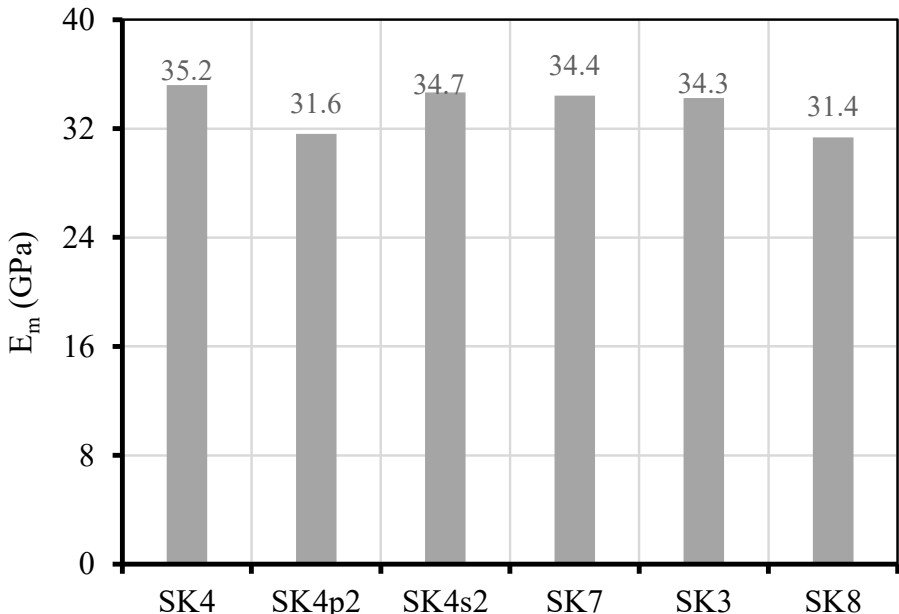

**Figure 9.** Young's modulus of eco-UHDC and eco-UHDFRC mixtures.

From the analysis of the above graph, it can be seen that: (1) The eco-UHDC with fibers (Sk4p2 and Sk4s2) had a lower Young's modulus; comparatively to the reference mixture, SK4, the difference is negligible in the mixture with steel fibers due to the small rise of air content. The biggest difference occurs with the eco-UHDFRC with polymeric fibers, Sk4p2, about 4 GPa less than the reference eco-UHDC, and the cause is the reduced stiffness of these fibers and the increase of air content (Table 4), being required to increase the superplasticizer dosage. (2) The SK3 mixture has a lower Young's modulus than the SK4 mixture, similarly to what happens with the compressive strength, contrary to the initial expectation, since SK3 has greater compactness. This result is probably due to the fact that this mixture does not have enough water to hydrate the cement and/or is due to the high dosage of superplasticizer used to obtain a fluidity similar to that of SK4. (3) The replacement of fly ash per pozzolan from Cape Verde, resulting in the mixture designated SK7, implied only a 2% decrease in the Young's modulus, leading to the conclusion that the two additions have approximately the same effect on the eco-UHDC mixture concerning Young's modulus, (4) The 10% increase of fly ash and the consequent reduction of limestone

filler resulted in a 10% decrease of the Young's modulus (comparing SK4 and SK8), similarly to what happened with the compressive strength. This is probably because the limestone filler greatly contributes to the hardening reaction with the fly ash, accelerating the process. Therefore, the increase of fly ash and the decrease of filler may have implied the loss of this beneficial effect since the fly ash only reacts much later. For more advanced ages, the difference tends to decrease, as can be seen in the compressive strength evolution (Figure 4).

### 3.2.2. Flexural Fracture Energy

The flexural fracture energy of the selected eco-UHDFRC was experimentally determined, and the force-displacement relation is presented in Figure 10. As expected, the cracks started in the notch and developed a multi-stage, elastic, post-crack, and post-peak hardening behavior with significant high deformation capacity.

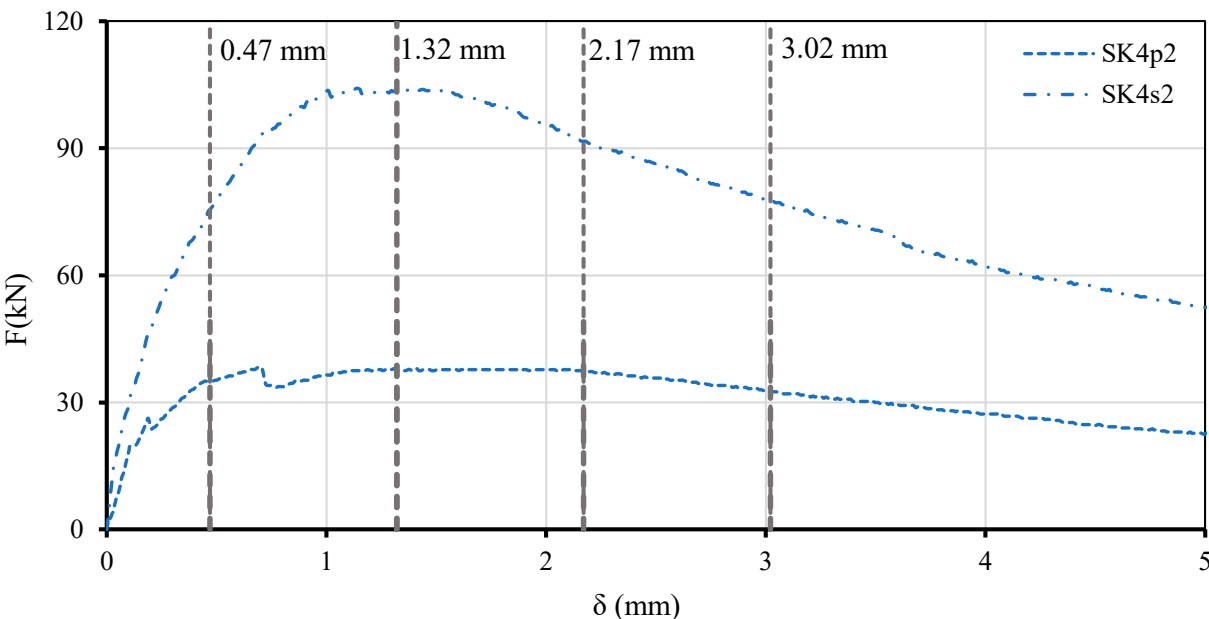

**Figure 10.** Load-deflection curves of the eco-UHDFRC residual flexural tensile strength test.

The residual flexural tensile strength that characterizes the behavior of the material up to a certain deflection ($f_{Ri}$), tensile stress under service conditions, and tensile stress used for checking ultimate limit states of structural elements produced with fiber concrete, $f_{Fts}$ and $f_{Ftu}$, were determined from Equations (1)–(3), respectively. The results are presented in Table 5.

$$f_{Ri} = \frac{3}{2} \times F_i \times 10^3 \times \frac{L_R}{b_R \times h_R{}^2} \tag{1}$$

$$f_{Fts} = 0.45 f_{R1} \tag{2}$$

$$f_{Ftu} = f_{Fts} - \frac{w_u}{CMOD_3}(f_{Fts} - 0.5 f_{R3} + 0.2 f_{R1}) \geq 0 \tag{3}$$

where $F_i$—force; $L_R$—distance between supports; $b_R$—width of specimen; $h_R$—distance between top of the notch and top of the cross section. The force $F_i$ is related to the deflection at mid-span, as follows: F1—force corresponding to a deflection of 0.47 mm; F2—force corresponding to a deflection of 1.32 mm; F3—force corresponding to a deflection of 2.17 mm; F4—force corresponding to a deflection of 3.02 mm; $CMOD_3$—crack mouth opening displacement corresponding to a deflection of 2.7 mm; and $w_u$ equal to $\delta_3$.

**Table 5.** Residual flexural tensile strength of eco-UHDFRC mixtures.

| | Residual Flexural Tensile Strength (MPa) | | | | | |
|---|---|---|---|---|---|---|
| **Mixtures** | $f_{R1}$ | $f_{R2}$ | $f_{R3}$ | $f_{R4}$ | $f_{Fts}$ | $f_{Ftu}$ |
| SK4s2 | 24 | 33 | 29 | 25 | 10.9 | 9.7 |
| SK4p2 | 11 | 12 | 12 | 11 | 5 | 4 |

*3.3. Shrinkage and Creep*

3.3.1. Shrinkage

The total shrinkage is the dimensional variation mainly caused by the combined effects of the drying shrinkage and the autogenous shrinkage. In UHPC, the autogenous shrinkage is the main component because this concrete has a very low water/binder ratio with extremely fine additions and because, generally, they do not include coarse aggregates [39,47,48].

The evolution of the shrinkage, $\varepsilon_{cs}$, measured in specimens produced with the select eco-UHDC is shown in Figure 11. Analyzing the graph in Figure 11, it is noted that, at very young ages, all mixtures have similar shrinkage. However, after 14 days, the differences became significant. The SK7, a mixture with pozzolan from Cape Verde, stands out for having the highest shrinkage with a difference approximately 15% higher than SK4, at 364 days. The SK8, with only 10% more fly ash than SK4, has the opposite behavior, presenting the lowest shrinkage because, as expected, the fly ash promotes the reduction of shrinkage. SK3, which differs from SK4 only in the lower W/B ratio, has a lower shrinkage than SK4 in certain phases, but then tends to stabilize and approach the values of SK4. A similar trend was observed by Eppers and Müller ([49] when the binder has a higher amount of C3A.

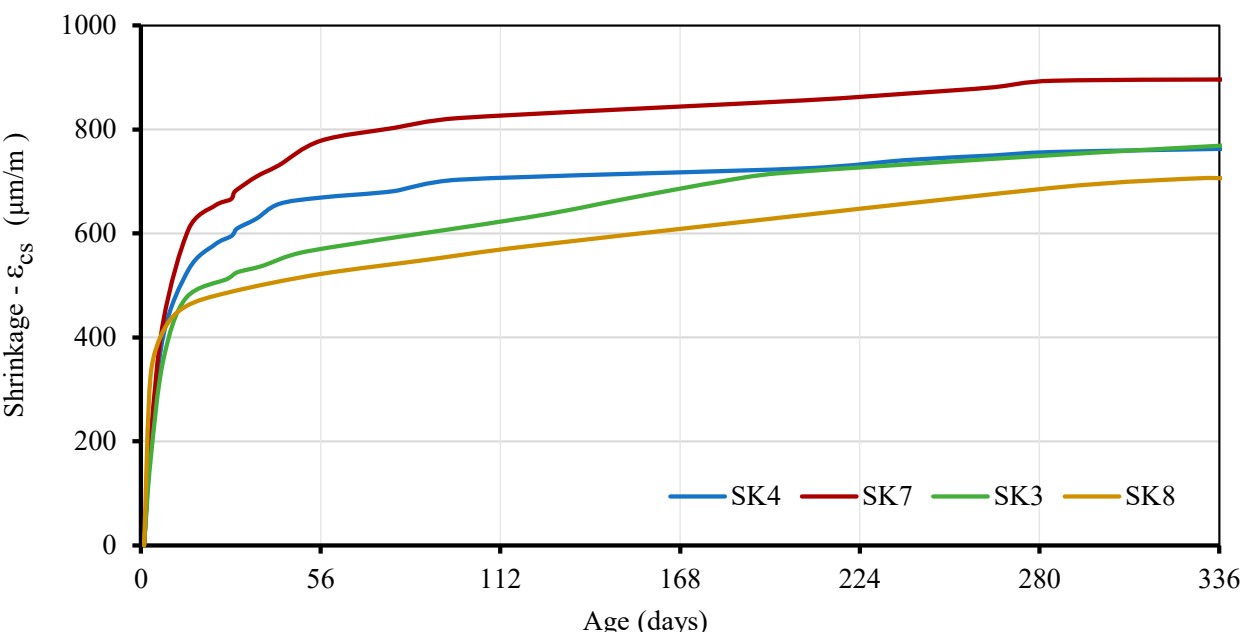

**Figure 11.** Evolution of total shrinkage of eco-UHDC with age.

Based on the results, it can be concluded that the partial replacement of cement by fly ash reduces the total shrinkage of the concrete, contrary to what happens with the replacement by pozzolan from Cape Verde. However, the results obtained in all characterized eco-UHDC are in line with the expected, since, according to Russel and Graybeal [50], the total shrinkage in UHPC is usually greater than 900 µm/m, depending on the curing method. Eppers and Müller [49] also reported autogenous shrinkage values of 600 to 900 µm/m at 28 days.

### 3.3.2. Creep

The obtained results for the creep characterization are presented through the evolution over time of the creep coefficient, $\varphi_c(t)$ (Figure 12), having the load applied at 28 days of age to assure a proper maturity of concrete. The $\varphi_c(t)$ is defined as the quotient between creep strain and elastic strain.

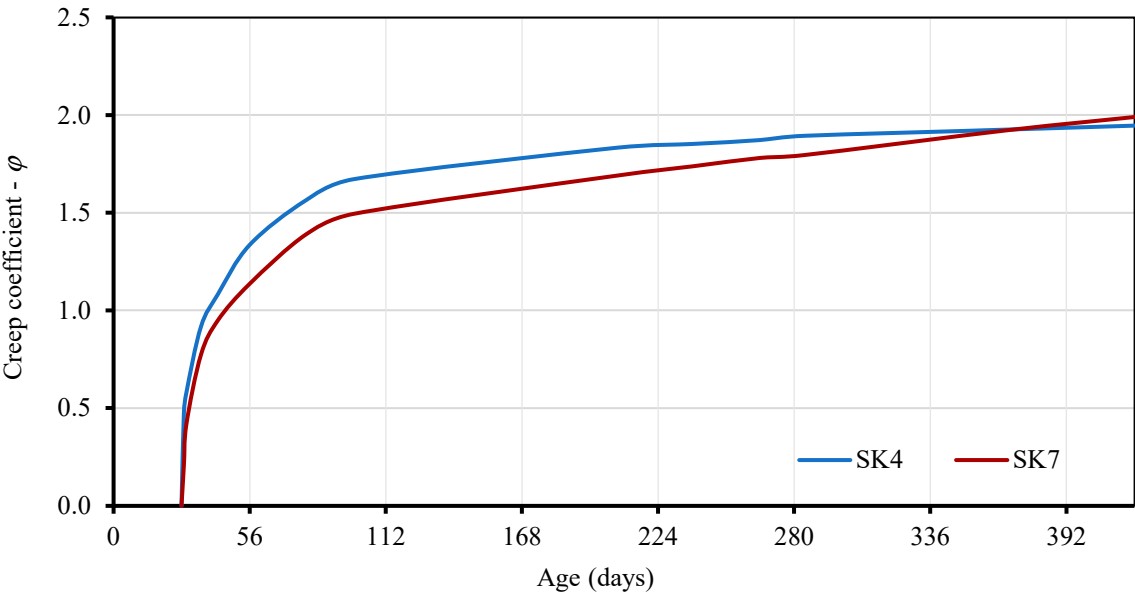

**Figure 12.** Evolution with age of creep coefficient: eco-UHDC with Cape Verde pozzolan (SK7) and eco-UHDC with fly ash (SK4).

Through the analysis of the curves presented in Figure 12, it can be concluded that the eco-UHDC with pozzolan from Cape Verde (SK7) initially has a creep coefficient about 13% lower than the eco-UHDC with fly ash (SK4). However, after one year, both mixtures show a similar creep coefficient and it is less than 2. It is also noted that the creep coefficient of SK4 seems to be stabilizing while the creep coefficient of SK7 appears to still evolve, which leads to the conclusion that pozzolan from Cape Verde tends to have higher creep coefficients than fly ash. The different maturity evolution of mixtures with these two pozzolanic additions is probably the reason for this effect. The reactivity of the Cape Verde pozzolan seems to be higher at younger ages and the fly ash has more pronounced evolution with age.

### 3.4. Durability—Exposed to Laboratory Environment

### 3.4.1. Carbonatation Resistance

The accelerated carbonation tests were carried out on SK4, SK7, Sk4p2, and SK4s2, and the average carbonation depth, $C_d$, is presented in Table 6. It is verified that, after 7 and 28 days of exposure, the carbonation in the eco-UHDC was null. The beginning of carbonation in the eco-UHDC started only at 56 days, being almost imperceptible. The two mixtures, SK4 and SK7, with fly ash and Cape Verde pozzolan, respectively, behaved similarly in all exposure periods.

**Table 6.** Carbonation depth in the eco-UHDC for various exposure periods.

| Mixtures | $C_{7\,Days}$ (mm) | $C_{28\,Days}$ (mm) | $C_{56\,Days}$ (mm) | $C_{90\,Days}$ (mm) |
| --- | --- | --- | --- | --- |
| SK4 | 0 | 0 | 2 | 2 |
| SK7 | 0 | 0 | 2 | 2 |
| SK4s2 | 0 | 0 | 0 | 2 |
| SK4p2 | 0 | 0 | 0 | 0 |

The incorporation of fibers indirectly improved the carbonation resistance of the eco-UHDC matrix, with greater emphasis of the polymeric fibers since the concrete with this type of fibers had no carbonation in any of the analyzed periods. Even in the mixture with steel fibers (SK4s2), the carbonation depth was only observed after 90 days of exposure. The concrete carbonation resistance mainly depends on the porosity of the exposed surface, as well as on the distribution of the connected pores, and this porosity is related to the cementitious matrix and its compactness. In general, this type of concrete has very low porosity. However, the addition of fibers in this type of matrices, in addition to the characteristics provided in the hardened state, promotes a more efficient mixing process. This is due to the combined action of the fibers with the paste viscosity, resulting in macro and micro mixing shear forces that make the fresh mixture more flowable, releasing more air content. For this reason, fiber-reinforced UHDC mixtures commonly exhibit slightly lower porosity than unreinforced UHDC mixtures, being more stable and better controlling the exudation of water in the fresh state. In the latter, the water exuded from the mixture can originate connected pores that slightly promote the penetration of aggressive agents, whereas, in the former, the pores are less connected. Moreover, the addition of fibers provides a reduction of micro-cracks caused by drying shrinkage.

### 3.4.2. Carbonatation Resistance

The results of capillary absorption with time $t_i$, $Sa(t_i)$, resulting from the average of three specimens for each eco-UHDC tested, are shown in Figure 13. The height of capillary in each specimen, $h(t_i)$, was obtained through the arithmetic mean of the four faces (Figure 13).

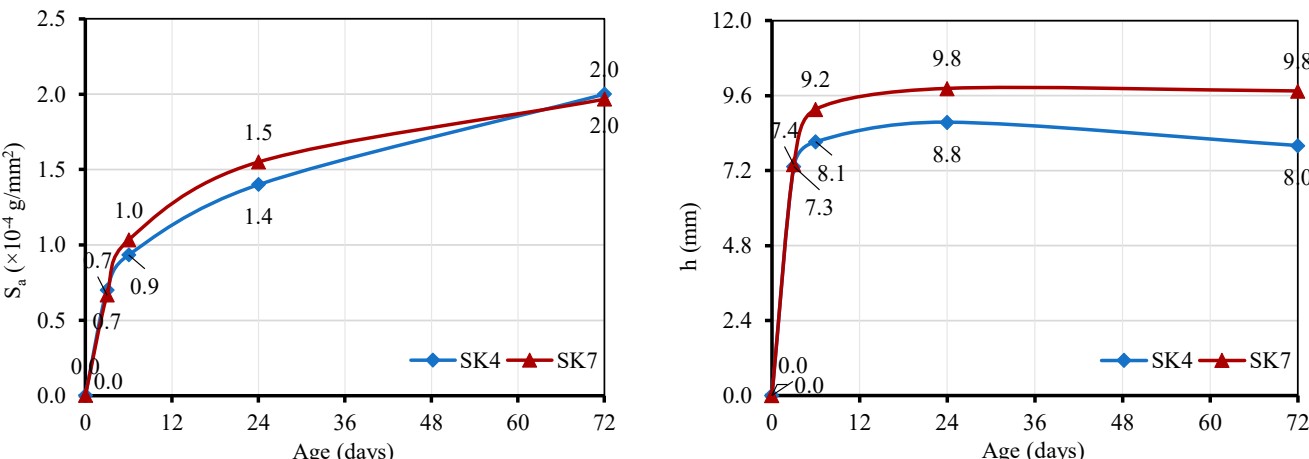

**Figure 13.** Results of capillary water absorption: capillary absorption, Sa (**left**); capillary ascension height (**right**).

The two eco-UHDCs clearly show a more pronounced capillary rise in the early stages, with a continuous increase up to 72 h. The eco-UHDC with pozzolan from Cape Verde presents, after 3 h, a capillary absorption slope slightly higher than that of the eco-UHDC with fly ash; however, after 72 h, the two concrete mixtures present similar results.

Figure 14 shows the relation between the capillary absorption, Sa, and the square root of time, obtained for each eco-UHDC as well as the trend line corresponding to the best fit of the points.

The trend lines of the two eco-UHDCs show correlation coefficients greater than 0.90. The slope of these lines represents the capillary absorption coefficient, and this coefficient is similar in the two concretes. This graph also shows the limit line corresponding to high-quality concrete, Sa = 0.1 mg/(mm$^2$·min$^{1/2}$), according to Browne [51]. This author proposed the following classification for the quality of concrete as a function of the Sa coefficient: above 0.2 mg/(mm$^2$·min$^{1/2}$) is "low-quality concrete"; between 0.1 and 0.2 mg/(mm$^2$·min$^{1/2}$) is "medium quality concrete"; and below 0.1 mg/(mm$^2$·min$^{1/2}$)

is "high-quality concrete". The two concretes can be classified as high-quality concretes. The results are far below the limit defined by the line Sa = 0.1 mg/(mm$^2$·min$^{1/2}$), proving through this durability indicator that both additions can be used as a partial substitute of cement without affecting this performance.

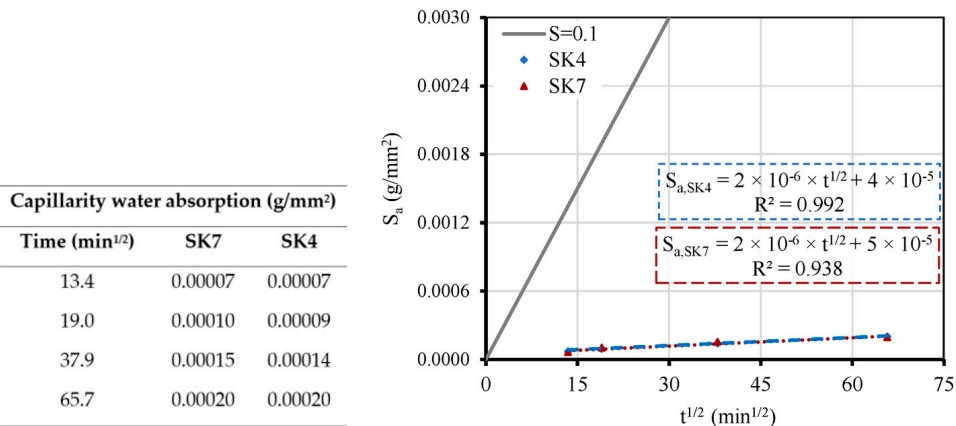

| Capillarity water absorption (g/mm²) | | |
| --- | --- | --- |
| Time (min$^{1/2}$) | SK7 | SK4 |
| 13.4 | 0.00007 | 0.00007 |
| 19.0 | 0.00010 | 0.00009 |
| 37.9 | 0.00015 | 0.00014 |
| 65.7 | 0.00020 | 0.00020 |

**Figure 14.** Capillarity water absorption in function of square root of time.

### 3.4.3. Chloride Migration Coefficient

The penetration depth of chlorides in specimens after the chloride migration test in a non-stationary regime was performed as shown in Figure 15.

Based on values of penetration depth, the non-steady-state chloride migration coefficient, $D_{nssm}$, was calculated using Equation (4).

$$Dnssm = \frac{0.0239\ (273+T)L}{(U-2)t} \times \left( x_d - 0.0238\sqrt{\frac{(273+T)\ L\ x_d}{U-2}} \right) \quad (4)$$

where U—absolute value of the applied voltage, V; T—average value of the initial and final temperatures in the anodic solution, °C; L—thickness of the specimen, mm; $x_d$—average value of the penetration depths, mm; and *t*-test—duration, hour.

Figure 16 shows the results of non-steady-state chloride migration coefficient, $D_{nssm}$, of the eco-UHDC mixtures obtained by the average of three specimens for each age. The same graphic shows the limits proposed by Luping [52] to classify concrete as a function of the chloride diffusion coefficient measured in the migration test. According to Luping [52], concrete is classified regarding the resistance to migration of chloride ions as: (i) very good strength concrete if $D_{nssm}$ is less than $2 \times 10^{-12}$ m$^2$/s; (ii) good resistance concrete if the $D_{nssm}$ is between $2 \times 10^{-12}$ m$^2$/s and $8 \times 10^{-12}$ m$^2$/s; (iii) moderate resistance concrete if $D_{nssm}$ is between $8 \times 10^{-12}$ m$^2$/s and $16 \times 10^{-12}$ m$^2$/s; and (iv) concrete not suitable for aggressive environments if $D_{nssm}$ value is greater than $16 \times 10^{-12}$ m$^2$/s. Therefore, the lower the $D_{nssm}$, the greater the resistance to chloride migration and, consequently, the greater the durability. All eco-UHDC mixtures correspond to good resistance concrete except eco-UHDC with Cape Verde pozzolan (SK7), which proves to be a very good resistance concrete at all ages. Figures 15 and 16 prove that mixtures improve resistance to chloride migration with age due to the continuous pozzolanic effect.

The mixture with pozzolan from Cape Verde (SK7) presents very low $D_{nssm}$ results compared to the other mixtures, with differences ranging between 27% and 56%. Comparing the SK4 and SK7, which differ only in the addition type, the $D_{nssm}$ results of SK7 are much lower, with differences of 56% at 28 and 56 days of age and 27% at 90 days (Figure 17a). This fact proves that the pozzolanic activity of fly ash is lower and slower than that of pozzolan from Cape Verde.

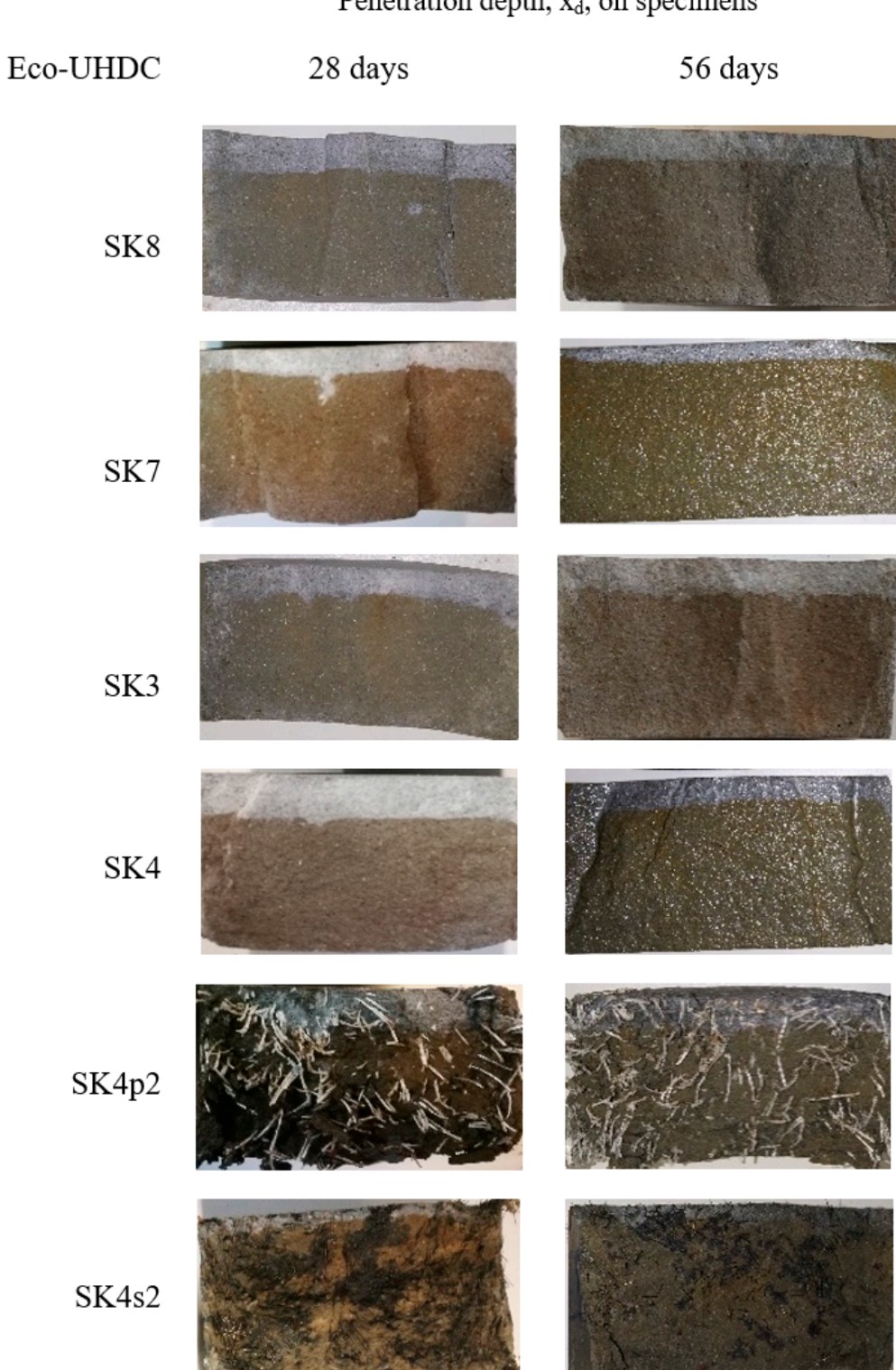

**Figure 15.** Depth of chloride penetration.

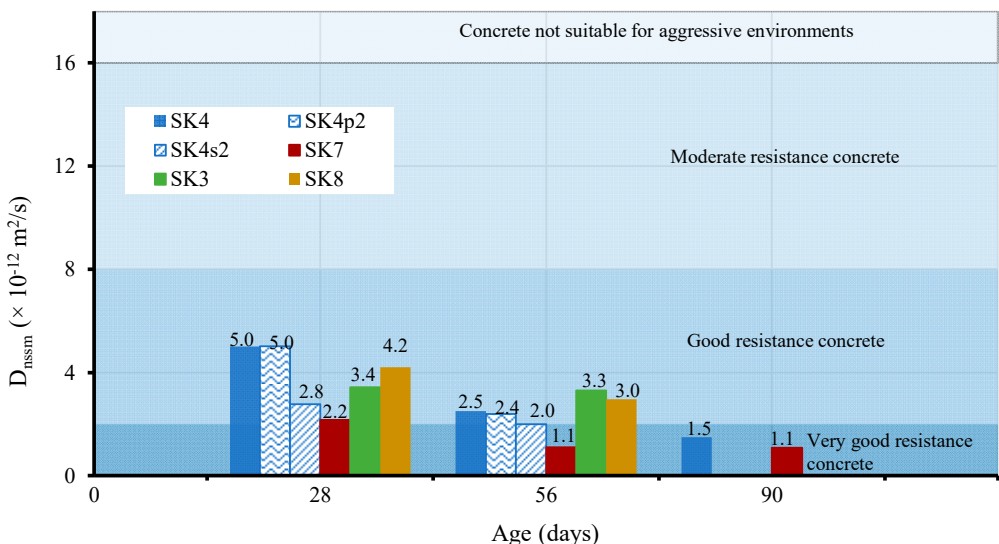

**Figure 16.** Non-steady state chloride migration coefficient, $D_{nssm}$, of the eco-UHDC mixtures at 28, 56, and 90 days.

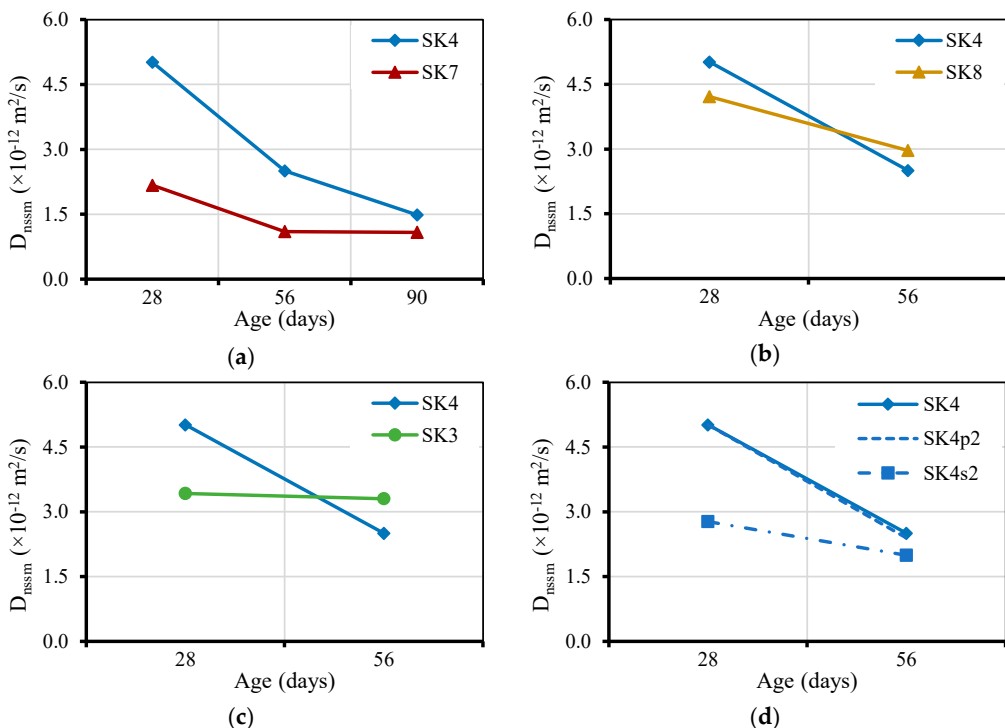

**Figure 17.** Variation of non-steady-state chloride migration coefficient with the age of eco-UHDC: (**a**) influence of Cape Verde pozzolan; (**b**) influence of the fly ash content, (**c**) influence of the W/B ratio, and (**d**) influence of the incorporation of fibers in the matrix.

Analyzing the influence of increasing the fly ash dosage, comparing the performance of SK4 and SK8 mixtures (Figure 17b), it is observed that the mixture with the highest dosage of fly ash has lower $D_{nssm}$ values at 28 days. However, at 56 days of age, the trend is reversed. This trend was not expected since the diffusion coefficient tends to decrease with the increase in the percentage of addition of fly ash and with the increase of the concrete age [53]. Real et al. [54] found that the diffusion coefficient increases with an increasing percentage of fly ash, but only at young ages. These results were justified by the short period of wet curing and by the young age of the concrete when the tests were carried out (28 days), considering that it is a binder matrix with pozzolanic additions. The same authors

recommended that chloride penetration resistance tests should be carried out for more advanced ages in concretes with additions whose pozzolanic reactions develop more slowly to correctly evaluate the real contribution of this type of addition. Several researchers report this fact [55,56], at least up to a certain percentage of replacement. According to some authors [57–59], there is an optimal content of fly ash that should be used to replace cement. This optimum content is related to the amount of C-H available in the mixture to satisfy the pozzolanic reaction developed by the fly ash. The diffusion coefficient for a given W/B ratio increases with the percentage of cement replacement by fly ash; however, the contribution of fly ash, namely in the refinement of the microstructure, is only relevant when the development of pozzolanic reactions is effective [58].

Comparing SK4 with SK3 (concrete with a lower water/binder ratio compared to SK4), observing Figure 17c, it can also be seen that, at 56 days of age, SK4 has a $D_{nssm}$ 20% lower than SK3. Normally, the diffusion coefficient tends to increase in less compact concretes, regardless of the type and dosage of the binder [58]. However, the observed trend may be explained by the higher water/binder ratios, which facilitate faster cement hydration, producing more C-H and providing a faster activation of fly ash.

The mixture with polymeric fibers (SK4p2) does not show significant improvement compared to the reference eco-UHDC (SK4), unlike the mixture with steel fibers (SK4s2) which has lower $D_{nssm}$ values of about 40% at 28 days and 20% at 56 days, compared to SK4 at the same age (Figure 17d).

### 3.4.4. Electrical resistivity

The resistance to chloride penetration was also evaluated through the electrical resistivity measured on the surface of the eco-UHDC to indirectly assess the ability of chloride ions to diffuse inside the concrete. Figure 18 shows the electrical resistivity results of the eco-UHDC mixtures, using the average of four readings per specimen and per age. As expected, the electrical resistivity increases with age due to the continuous hydration of cementitious materials, mainly by the pozzolanic reaction, producing a finer pore network with less connectivity.

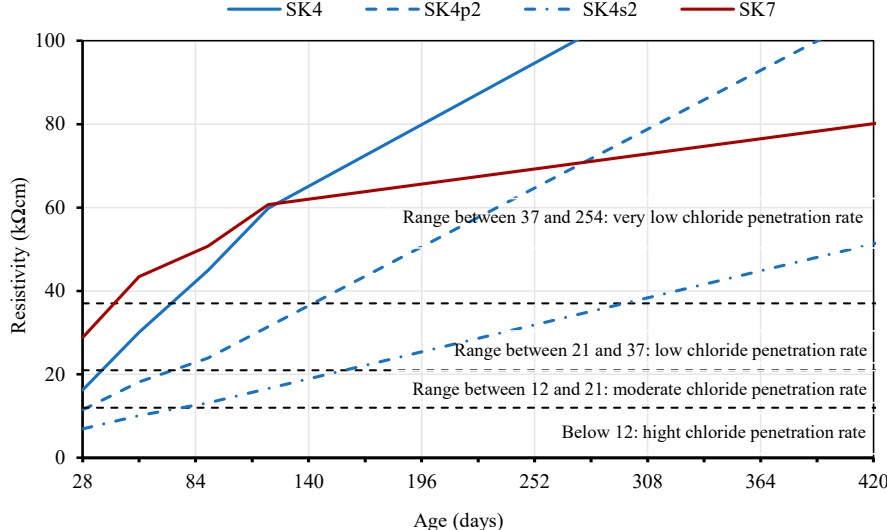

**Figure 18.** Variation of electrical resistivity of eco UHDC with age.

Comparing the electrical resistivity results of the developed eco-UHDCs, with the reference values proposed by AASHTO T358-15 [60], presented in Figure 18, it appears that the eco-UHDCs have a very low chloride penetration rate, but only after a long curing period (about 90 days for fiber-free concrete, 150 days for polymer fiber concrete, and 250 days for steel fiber concrete). This late effect is certainly due to the pozzolanic effect in the development of new C-S-H compounds.

The average electrical resistivity of SK7 (UHDC with Cape Verde pozzolan) is higher than that of SK4 (UHDC with fly ash) up to 90 days. From this age onward, there was a contrary trend, confirming the trend recorded in the study with LCC-low carbon concrete [61]. Comparing these values to those obtained in the chloride migration test in a non-stationary regime, this trend was not registered at the corresponding ages, although the graph presented in Figure 17a shows a convergence of results of the two concretes. This occurred probably because the prolonged curing period of the specimens submitted to the resistivity test allows the fly ash to have the necessary time to develop the pozzolanic reactions and, thus, improve the concrete performance.

The results obtained also show that the eco-UHDC with fibers, mainly with steel fibers, present a higher chloride penetration rate than the reference eco-UHDC, contradicting the results obtained in the chloride migration test in a non-stationary regime. These results corroborate what was reported in previous studies, that is, the presence of steel fibers influences the electric field generated by the resistivity device [62,63]. Therefore, when interpreting the results, it is necessary to bear in mind that it is not appropriate to directly compare the results of concrete with and without steel fibers.

### 3.4.5. Chloride Content at the Eco-UHDC Surface

Figure 19 shows the mean values of chloride content, Cs, per mass of cement and addition type II at 28, 56, and 90 days of age. These values were measured, initially, in concrete mass using Equation (5) and later calculated relatively to the mass of cement and type II addition using Equation (6) to be compared with the critical value of EN 206-1 [64]. One must consider that there is, in 100 g of SK4 mass, 35.8 g of cement plus fly ash, and in the case of SK7, 36.1 g of cement plus pozzolan from Cape Verde.

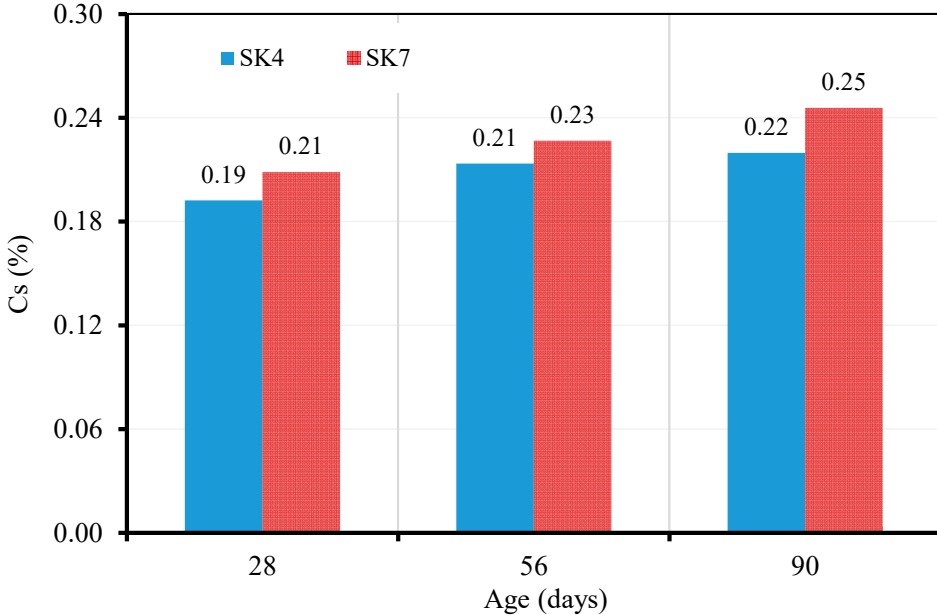

**Figure 19.** Chloride content at the eco-UHDC surface at 28, 56, and 90 days of age.

$$Cs = 3.545 \times f_{AgNO_3} \frac{V_{NH\,SCN}^{blanc} - V_{NHSCN}^{concrete}}{m_{concrete}} \tag{5}$$

$f_{AgNO_3}$—is molarity of silver nitrate solution;

$V_{NH\,SCN}^{blanc}$—volume of the ammonium thiocyanate solution used in the blank titration (mL);

$V_{NHSCN}^{concrete}$—volume of the ammonium thiocyanate solution used in the titration (mL);

$m_{concrete}$—mass of the concrete sample (g).

$$Cs_{m_{cement+addition\ type\ II}} = \frac{Cs}{m_{cement+type\ II\ addition}} \times 100 \qquad (6)$$

The results show that, in general, the eco-UHDC with pozzolans from Cape Verde has a higher content of chlorides on the surface. This result was already expected since this concrete has a lower diffusion coefficient of chloride ions, thus demonstrating that pozzolan provides a greater ability to fix chloride ions in concrete compared to fly ash. So, this ability potentiates a higher concentration of Cs because this resistance creates a barrier to chloride diffusion. However, the two concrete mixtures present values that range from 38 to 52% lower than the critical chloride content (% CR), whose value is 0.4% according to EN 206-1 [64] Considering these values, it is concluded that the probability of corrosion occurring at the end of the periods of exposure to chlorides to which these concretes were subjected is very low (according to LNEC specification E 463 [65]).

*3.5. Durability—Exposed to Maritime Environment*

3.5.1. Chloride Content at the Eco-UHDC Surface

Figure 20 shows the average values of Cs per mass of cement and the type II addition of eco-UHDC exposed to the environmental conditions of Cape Verde in the tidal zone for circa 30 months.

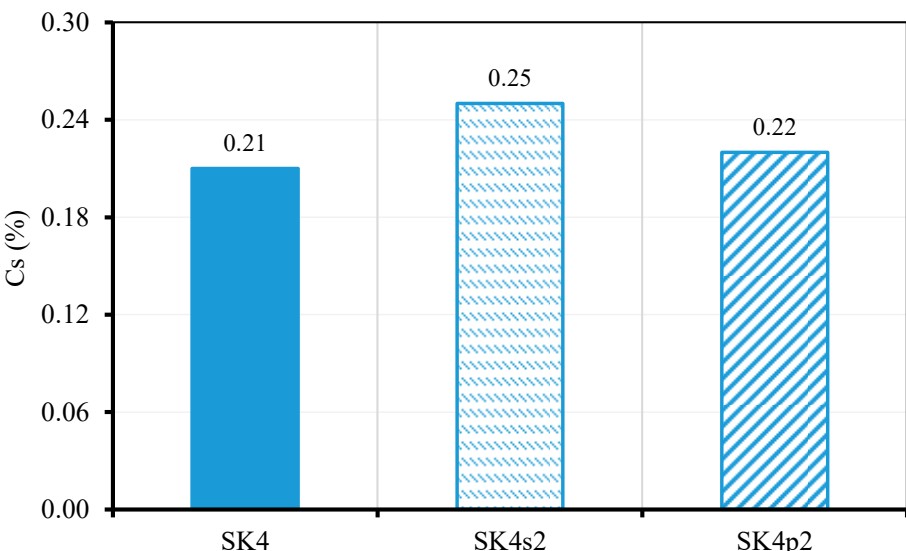

**Figure 20.** Chloride content per mass of cement on the surface of SK4, SK4s2, and SK4p2 concrete exposed for circa 30 months in the tidal zone.

The chloride content measured in SK4 and tested in a laboratory, when compared with the results of specimens exposed in the tidal zone, shows similar results. After circa 30 months of exposure to chlorides in the tidal zone, the results show that the probability of corrosion in the reinforcement of structures produced with the developed UHDC is almost nil, at least during the analyzed period since the Cs values are lower than the critical chloride content (% CR), which is, according to EN 206-1 [64], 0.4% (concrete with steel reinforcements or other embedded metals). The specimens with fibers present Cs values slightly higher than the reference concrete (SK4), approximately 18% higher in the SK4s2 (with steel fibers), and 6% higher in the SK4p2 (with polymeric fibers).

3.5.2. Depth of Chloride Ion Penetration

Figure 21 and Table 7 show the depth of chloride ion penetration obtained on each of the faces of the specimens. The depth of chloride ion penetration of each edge corresponds to the average of measurements at three points. There are some differences comparatively to the values measured in tests carried out following the LNEC E 463 [65]. The eco-UHDC

without fibers (SK4) has the lowest depth of chloride ion penetration, followed by the concrete with steel fibers (SK4s2), and, finally, the concrete with polymeric fibers (SK4p2). These results contradict the results obtained in the tests under laboratory conditions, where the eco-UHDC with steel fibers was the concrete with the lowest depth of chloride penetration. These differences may be related to sea salinity that are different from the laboratory conditions and to the wet/dry cycles of the tidal zone that are not reproduced in the laboratory.

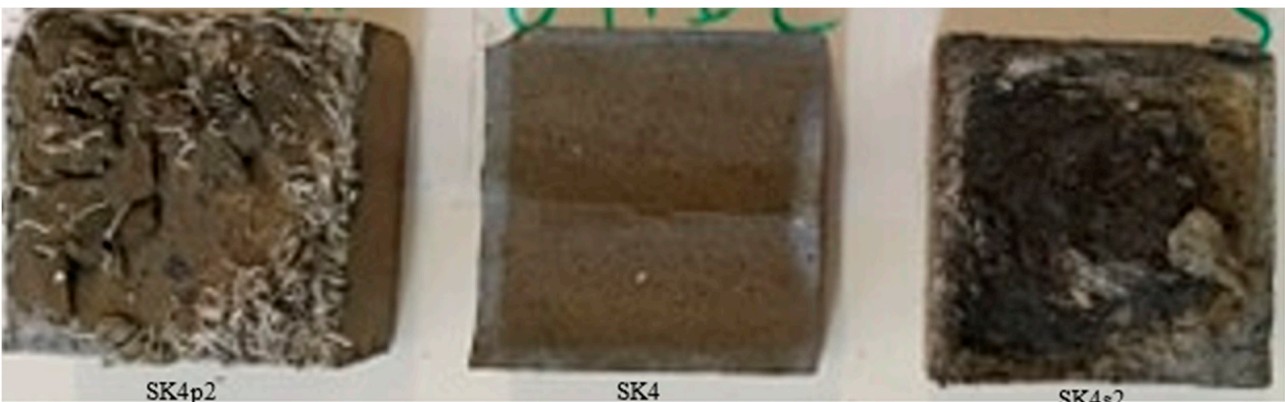

**Figure 21.** Depth of chloride ions penetration in eco-UHDC concrete exposed for circa 30 months in the tidal zone.

**Table 7.** Depth of chloride ions penetration, in mm, in UHDC concrete exposed for circa 30 months in the tidal zone.

| Eco-UHDC | SK4 | SK4s2 | SK4p2 |
|---|---|---|---|
| | **(mm)** | | |
| Edge A | 8 | 10 | 8 |
| Edge B | 7 | 7 | 15 |
| Edge C | 5 | 7 | 7 |
| Edge D | 4 | 7 | 5 |
| Average | 6 | 8 | 9 |

*3.6. Lifetime Expected and Minimum Cover Required*

The lifetime expected for a concrete structure is defined during the design phase and corresponds to the period during which the structure meets the safety, functionality, and aesthetic requirements without unforeseen maintenance costs [66]. These requirements are influenced not only by the mechanical and deferred properties of the concrete but also by the durability parameters. Most reinforced concrete structures are designed for a service lifetime of 50 or 100 years regarding the risk of rebar corrosion.

This section presents the expected service lifetime of reinforced concrete structures produced with the developed eco-UHDC concretes determined considering the resistances to carbonation and to the chlorides diffusion and the minimum cover defined in main codes. The estimated minimum concrete cover, $c_{min, dur}$, necessary to guarantee the proper corrosion resistance during the defined service lifetime for structures exposed to environmental conditions XC (corrosion induced by carbonation) and XS (induced by chloride ions present in seawater) are also presented. The results presented were obtained using the Tuutti model [67] for the degradation of reinforced concrete caused by steel corrosion, the recommendations described in the LNEC E465 [68], and considering two types of structures: (i) current structures with an intended service lifetime ($t_g$) equal to 50-years-old with a reliability class RC2 (service lifetime category or structural class S4), which corresponds to a safety factor $\gamma$ equal to 2.3 [68]; (ii) special structures with a $t_g$ equal to 100-years-old

with a reliability class RC3 (lifetime category or structural class S5), which corresponds to a safety factor $\gamma$ equal to 2.8 [68].

3.6.1. Prediction of the Service Lifetime of Reinforced Concrete Structures under the Environmental Exposure Classes XC

The prediction service lifetime for current and special structures produced with eco-UHDC, SK4, SK7, SK4s2, and SK4p2 exposed to XC environmental conditions are presented in Table 8. The developed concretes present resistance to carbonation, more than enough for the XC exposure class, and in all cases the estimated service life is over 400 years.

**Table 8.** Service lifetime (years) for current and special structures produced with the studied concrete under environmental conditions XC.

| | | Current Structures (Class S4, RC2, 50 Years) | | | | Special Structures (Class S5, RC3, 100 Years) | | | |
|---|---|---|---|---|---|---|---|---|---|
| **Exposure Class** | | XC2 | XC3 | XC4 * | XC4 ** | XC2 | XC3 | XC4 * | XC4 ** |
| **Eco-UHDC** | $C_{min,dur}$ (mm) EC 2 | 25 | | 30 | | 30 | | 35 | |
| SK4 | | >>400 | >>400 | >>400 | >>400 | >>400 | >>400 | >>400 | >>400 |
| SK7 | | >>400 | >>400 | >>400 | >>400 | >>400 | >>400 | >>400 | >>400 |
| SK4s2 | tg (years) | >>400 | >>400 | >>400 | >>400 | >>400 | >>400 | >>400 | >>400 |
| SK4p2 | | >>400 | >>400 | >>400 | >>400 | >>400 | >>400 | >>400 | >>400 |

XC4 *—dry region; and XC4 **—Wet region.

3.6.2. Determination of the Minimum Cover to Resist Carbonation-Induced Corrosion

The minimum cover required to guarantee the resistance against steel corrosion in reinforced concrete structures under the environmental exposure classes XC are presented in Table 9. These covers are much lower than the minimum values recommended by Eurocode 2 (EC2) [69] and EN 206-1 [64], with differences greater than 20 mm. In other words, the results showed that, for any structural class, the structures produced with eco-UHDC can have a cover less than 5 mm for any class of XC environmental exposure, considering only the criterion of durability. These results have a significant impact on the cost of the structures since it will be possible to produce elements with smaller cross-sections. It can also be seen that the eco-UHDC with fly ash (SK4) and the eco-UHDC with pozzolan from Cape Verde (SK7) have similar minimum covers. The introduction of fibers in eco-UHDC (SK4s2 and SK4p2) allows reducing the cover, the highest difference was obtained with the concrete with polymeric fibers (SK4p2); the minimum cover was reduced by half.

**Table 9.** Cover for S4 and S5 class structures under environmental conditions XC: minimum standards cover and minimum covers for the studied concrete.

| | | | Current Structures (Class S4, RC2, 50 Years) | | | | Special Structures (Class S5, RC3, 100 Years) | | | |
|---|---|---|---|---|---|---|---|---|---|---|
| **Exposure Class** | | | XC2 | XC3 | XC4 * | XC4 ** | XC2 | XC3 | XC4 * | XC4 ** |
| | | EC2 | 25 | 25 | 30 | | 30 | 30 | 35 | |
| Minimum cover $C_{min,dur}$ (mm) | Eco-UHDC | SK4 | 1.3 | 1.9 | 2.6 | 2.9 | 1.7 | 3.0 | 4.0 | 4.2 |
| | | SK7 | 1.3 | 1.9 | 2.6 | 2.9 | 1.7 | 3.0 | 4.0 | 4.2 |
| | | SK4s2 | 1.2 | 1.7 | 2.4 | 2.6 | 1.6 | 2.7 | 3.6 | 3.8 |
| | | SK4p2 | 0.6 | 0.9 | 1.2 | 1.3 | 0.8 | 1.3 | 1.8 | 1.9 |

XC4 *– dry region; and XC4 **– Wet region.

### 3.6.3. Prediction of the Service Lifetime of Reinforced Concrete Structures under the Environmental Exposure Classes XS

The prediction service lifetime of current and special structures exposed to the XS environmental conditions, produced with the developed eco-UHDC, are presented in Table 10. The results show that, adopting the minimum cover recommended by Eurocode 2 (EC2) [69] and EN 206-1 [64], the service lifetime of structures produced with the eco-UHDC developed is adequate and significant when exposed to the environmental conditions XS1 and XS2. It is recalled that current structures and special structures must have a service lifetime of at least 50 and 100 years, respectively. These results may indicate some long-term economic and environmental benefits because the need for maintenance will occur later than expected. The SK7 mixture proves the higher resistance to chloride diffusion, fulfilling the required service life for all exposure classes (including XS3) and both structural classes.

**Table 10.** Service lifetime (years) for current and special structures produced with the developed concrete under environmental conditions XS.

| | | | Current Structures (Class S4, RC2, 50 Years) | | | | Special Structures (Class S5, RC3, 100 Years) | | | |
|---|---|---|---|---|---|---|---|---|---|---|
| Exposure Class | | | XS1 | XS2 | | XS3 | XS1 | XS2 | | XS3 |
| | | | | 1 m | 1.4–25 m | | | 1 m | 1.4–25 m | |
| | SK3 | | 573 | 115 | 91 | 41 | 852 | 174 | 145 | 54 |
| | SK4 | | 246 | 77 | 66 | 18 | 365 | 127 | 112 | 23 |
| Eco-UHDC | SK7 | $t_g$ | 1581 | 211 | 158 | 113 | 2357 | 296 | 229 | 148 |
| | SK8 | (years) | 362 | 91 | 75 | 26 | 539 | 145 | 124 | 34 |
| | SK4s2 | | 915 | 150 | 115 | 65 | 1361 | 218 | 175 | 86 |
| | SK4p2 | | 246 | 77 | 66 | 17 | 365 | 127 | 112 | 23 |

### 3.6.4. Minimum Cover to Resist Corrosion Induced by Chloride Ions Present in Seawater

The minimum cover required to guarantee the resistance against steel corrosion induced by chlorides in reinforced structures produced with the eco-UHDC are presented in Table 11. Comparing the minimum cover required in the structures produced with the eco-UHDC with the minimum covers recommended by Eurocode 2 (EC2) [69] it is possible to conclude that the developed eco-UHDCs provide good corrosion protection because the minimum cover required is lower than the values recommended by codes for exposure classes XS1 and XS2, with differences ranging from 12% to 55%, depending on the type concrete and exposure class. However, for current structures and exposure class XS3 (tidal zones), only the eco-UHDC with pozzolan from Cape Verde (SK7) and the eco-UHDC with steel fibers (SK4s2) proved to be suitable, since they are the only ones with a required cover below the minimum recommended by codes, with differences of 8 mm and 4 mm, respectively. It is concluded, once again, that the concrete with pozzolan from Cape Verde is more suitable to resist the penetration of chloride ions than the concrete with fly ash.

**Table 11.** Cover for S4 and S5 class structures under environmental conditions XS: minimum standards cover and minimum covers for the studied concrete.

| Exposure Class [1] | | | Current Structures (Class S4, RC2, 50 Years) | | | | Special Structures (Class S5, RC3, 100 Years) | | | |
|---|---|---|---|---|---|---|---|---|---|---|
| | | | XS1 [2] | XS2 [3] | | XS3 [4] | XS1 [2] | XS2 [3] | | XS3 [4] |
| | | | | 1 m | 1.4–25 m | | | 1 m | 1.4–25 m | |
| Minimum cover $C_{min,dur}$ (mm) | | EC2 | 35 | 40 | | 45 | 40 | 45 | | 50 |
| | Eco-UHDC | SK3 | 20 | 23 | 26 | 47 | 25 | 29 | 33 | 58 |
| | | SK4 | 24 | 28 | 31 | 57 | 30 | 36 | 39 | 70 |
| | | SK7 | 16 | 18 | 20 | 37 | 20 | 23 | 26 | 46 |
| | | SK8 | 22 | 26 | 29 | 52 | 27 | 33 | 36 | 64 |
| | | SK4s2 | 18 | 21 | 23 | 42 | 22 | 26 | 29 | 52 |
| | | SK4p2 | 24 | 28 | 31 | 57 | 30 | 36 | 39 | 70 |

[1] In this analysis, regarding the distance to the coastline, the worst-case scenario was considered (structures are located next to the coastline); [2] Structures exposed to the air with sea salts; [3] Structures permanently submerged; [4] Structures in the tidal zone.

## 4. Conclusions

The UHDC matrix was optimized using a combination of filler material (limestone or quartz flour) and reactive pozzolanic additions as a partial replacement of Portland cement, aiming to improve its eco-efficiency. Various types of fibers (steel, polymeric, glass, and basalt) were also used to enhance the mechanical and durability performance. Based on the results experimentally obtained, the following conclusion are drawn:

(1) The optimization of the unreinforced UHDC matrix makes it possible to produce concrete with only 60% of cement in relation to the total binder, maintaining good workability and the desired mechanical characteristics (compressive strength higher than 100 MPa and flexural strength higher than 12 MPa at 56 days of age). The eco-UHDC matrix with fly ash develops greater compressive strength, up to 25%, compared to the eco-UHDC with other additions;

(2) The eco-UHDC with pozzolan from Cape Verde and fly ash show similar carbonation resistances and creep coefficients (lower than 2.0), but the shrinkage of the former is 15% higher than that of the latter at 364 days. It must also be highlighted that pozzolan from Cape Verde has a huge effect in reducing the diffusion coefficient of chloride ions, reaching 57% lower than that of the eco-UHDC with fly ash, but the difference decreases with the age of concrete due to the slower hardening provided by the fly ash;

(3) The shrinkage of all developed eco-UHDC varies between 700 and 900 μm/m after stabilization. This is an expected outcome since the total shrinkage of UHPC is normally higher than 900 μm/m and replacing cement with fly ash decreases the total shrinkage of concrete. The eco-UHDC with the replacement of cement by natural pozzolan from Cape Verde has an opposite trend;

(4) The addition of steel fibers and polymeric fibers provides greater ductility comparatively to the addition of glass and basalt fibers. Steel fibers obviously increase the flexural tensile strength, between 50 and 176%, depending on the fiber addition rate. The other types of fibers also have an influence on tensile strength, but their dosage must be equal to or higher than 2% volume. The glass and polymeric fibers have much more of a relevant effect than basalt fibers in the mechanical characterization;

(5) The introduction of fibers in eco-UHDC affects in different ways some mechanical properties. While the addition of steel fibers can increase up to 27% of the compressive strength, the basalt and glass fibers can decrease to circa 10% in both the compressive strength and the Young's modulus. These losses are due to the negative effect of those fibers on the workability and air release of the matrix, resulting in less compact mixtures and, thus, with higher air content, less stiffness, and less strength;

(6) The addition of steel and polymeric fibers also improves the durability performance of eco-UHDC since there is an increase in resistance to both carbonation and chloride

ions. Polymeric fibers are the most effective in terms of carbonation resistance since the UHDC with this type of fibers exhibited no carbonation in any of the analyzed periods. The UHDC reinforced with steel fibers is more suitable to be used in an environment under the action of chloride ions because this mixture reduces the migration coefficient, $D_{nssm}$, by 40% at 28 days and 20% at 56 days in comparison with the eco-UHDC used as reference (SK4);

(7) Considering only the requirements related to durability, the structures produced with the developed eco-UHDC, exposed to environmental conditions XC, XS1, and XS2, require a cover lower than the values recommended by EC2, with differences that can reach 55%. From all the developed eco-UHDC, the one with pozzolana from Cape Verde is the most suitable to be used in any structural class, submitted to any XC and XS exposure classes.

These results prove that it is possible to produce structural elements with eco-UHDC, with reduced cross-sections and high durability, and consequently, with economic and environmental benefits. The topic of concrete sustainability associated with the cost and energy consumption required to produce this type of concrete will be addressed in future research.

**Author Contributions:** K.R.: Conceptualization, methodology, formal analysis, writing—original draft preparation. H.C.: Conceptualization, methodology, validation, supervision, writing—review and editing. R.C.: Conceptualization, validation, supervision, formal analysis, writing—review and editing. E.J.: Conceptualization, supervision, writing—review and editing. All authors have read and agreed to the published version of the manuscript.

**Funding:** This research was funded by [Calouste Gulbenkian Foundation] grant number [SBG35-2016] and by [FCT] in the framework of project [UIDB/04625/2020].

**Institutional Review Board Statement:** Not applicable.

**Informed Consent Statement:** Not applicable.

**Data Availability Statement:** Data will be made available on request.

**Acknowledgments:** The authors acknowledge the Calouste Gulbenkian Foundation for the Doctoral Grant with the reference SBG35-2016; the ISEC—Polytechnic Institute of Coimbra for providing the facilities and all necessary resources to perform the study; and the companies Secil, Omya, Sarendur and BASF for their support. This work is part of the research activity carried out at Civil Engineering Research and Innovation for Sustainability (CERIS) and has been funded by Fundação para a Ciência e a Tecnologia (FCT) in the framework of project UIDB/04625/2020.

**Conflicts of Interest:** The authors declare no conflict of interest.

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
