# Peer review of "Development and Characterization of Eco-Efficient Ultra-High Durability Concrete"

_sustainability, doi:10.3390/su15032381_

Round 1

Reviewer 1 Report

Tables and text in the document needs to be better formatted.,

When tables are called up, indicate the table number which the content is referring to.  Example in paragraph 2 in section 2.2. 

Why were the results of strengths at 7 days presented? 

Behaviours such as what was presented in figure 8 was not concluded satisfactorily in the conclusion section. Conclusion should completely summarize the discussion. 

Author Response

The authors would like to acknowledge the reviewers for the detailed revision of the paper. Revisions were made considering the reviewers amendments and suggestions. The authors believe these changes greatly increased the manuscript quality.

A detailed responses to the reviewer’s questions are provided point by point. Please see the attachment.

Reviewer 2 Report

The paper describes an extensive study on eco-efficient ultra-high durability concrete.  Results of fresh, hardened and durability material performance are presented.  A few commons are made, based on the current version of manuscript.

1. The technical advancement in the formation of eco-UHDC mixtures is achieved by optimizing the mixture proportion.  However, from the view of material development, the research innovation is not very high, and the design method is not rare.

2. The main parts of the research are about testing of mechanical properties and durability for eco-UHDC.  However, the topic of concrete sustainability is also associated with the cost and energy consumption when producing this type of concrete.  The comparation of eco-UHDC and conventional concrete in terms of waste, energy, and cost estimation is also important.

3. In section 3.4.1 and Table 4, “The incorporation of fibers significantly improved the carbonation resistance of the eco-UHDC matrix, with greater emphasis of the polymeric fibers since the concrete with this type fibers had no carbonation in any of the analyzed periods.”, this is not convincing because the fibers should play role in crack resistance rather than carbonation resistance.  In addition, the discussion is suggested to be verified by measuring the porosity of material.

4. The presentation could be more concise.  The current manuscript gives a sense that it is like a long technical report, not a journal article.

Author Response

(The authors gave the same response as above.)

Reviewer 3 Report

The objective of this manuscript is to develop eco-ultra-high durability concrete (eco-UHDC), optimizing the UHDC matrix, focusing mainly on durability and seeking the lowest environmental impact, using a combination of filler material and reactive pozzolanic additions as partial replacement of Portland cement, and also using various types of fibers to improve performance. In summary, the research is exciting and provides valuable results. Still, the current manuscript has several weaknesses that must be strengthened to obtain a documentary result equal to the publication’s value.

The abstract is complete and well-structured but needs to explain the contents of the manuscript better, and the authors need an in-depth study of the quantitative results of this research.

Scientific innovation is limited in the introduction of the manuscript. The authors must go deeper and detail how this research differs from countless others on this topic, and this must be evidenced together with the objectives at the end of the introduction.

The first paragraph introducing the research topic may present a much broad and comprehensive view of the problems related to your topic with citations to authority references  (Bond behaviors of pre- and post-yield deformed rebar embedded in ultra-high performance concrete. Construction and Building Materials 2022). 

The expression of Table. 2 and Table. 3 on Page 6 needs to be clarified and should be revised. And the serial number of the table on Page 17 is mislabeled. Similar problems in the whole text should be checked and addressed carefully.

A large part of this manuscript has formatting errors, which is not enough for a paper, and should be revised carefully.

The conclusions of the manuscript do not provide a good summary of the results of the study, and it should mention the scope for further research as well as the application of the study.

Author Response

(The authors gave the same response as above.)

Author Response

(The authors gave the same response as above.)

Round 2

Reviewer 2 Report

The authors have well responded to the reviewer's comments.  However, the presentation can be further improved.  For example, add unit "mm"? in Table 7.

Author Response

Answers to Reviewers_round 2

The authors would like to thank the reviewers for all suggestions and also for the positive comments.

Reviewer 2

The authors have well responded to the reviewer's comments.  However, the presentation can be further improved.  For example, add unit "mm"? in Table 7.

Authors reply:

The units “mm” were added to Table 7. Thank you.

The formatting of the document as well as the overall presentation has been checked in order to eliminate any typos and also to improve the presentation.

Reviewer 4

The authors have provided satisfactory answers to the questions raised, therefore manuscript can be accepted in its present form.

Authors reply:

Thank you.

Reviewer 4 Report

The authors have provided satisfactory answers to the questions raised, therefore manuscript can be accepted in its present form

Author Response

(The authors gave the same response as above.)
